

# Experimental study on retardation of a heavy NAPL vapor in partially saturated porous media

Simon M. Kleinknecht[1], Holger Class[2], and Jürgen Braun[1]

[1]Facility for Subsurface Remediation, Institute for Modelling Hydraulic and Environmental Systems, University of Stuttgart, Pfaffendwaldring 61, 70569 Stuttgart, Germany
[2]Department of Hydromechanics and Modelling of Hydrosystems, Institute for Modelling Hydraulic and Environmental Systems, University of Stuttgart, Pfaffendwaldring 61, 70569 Stuttgart, Germany

*Correspondence to:* Simon M. Kleinknecht (simon.kleinknecht@iws.uni-stuttgart.de)

**Abstract.** NAPL contaminants introduced into the unsaturated zone spread as a liquid phase; however, they can also vaporize and migrate in a gaseous state. Heavy vapors preferentially migrate downward due to their greater density and, thus, pose a potential threat to underlying aquifers. Large-scale column experiments were performed to quantify partitioning processes responsible for the retardation of carbon disulfide ($CS_2$) vapor in partially saturated porous media. The results were compared

with a theoretical approach taking into account the partitioning into the aqueous phase. The experiments were conducted in large, vertical columns (i.d. = 0.109 m) of 2 m length packed with different porous media. A slug of $CS_2$ vapor and the conservative tracer argon was injected at the bottom of the column followed by a nitrogen chase. Different seepage velocities were applied to characterize the transport and to evaluate their impact on retardation. Concentrations of $CS_2$ and argon were measured at the top outlet of the column using two gas chromatographs. The temporal-moment analysis for step input was

employed to evaluate concentration breakthrough curves and to quantify diffusion/dispersion and retardation. The experiments conducted showed a pronounced retardation of $CS_2$ in moist porous media as a function of porous medium and water saturation. An increase in the retardation coefficient with increasing water saturation was observed. Thus, the novel vapor-retardation experiments demonstrated that migrating $CS_2$ vapor is retarded as a result of partitioning into the aqueous phase. Moreover, $CS_2$ which is dissolved in the pore water is amenable to biodegradation. First evidence of $CS_2$ decay by biodegradation was

found in the experiments. The findings contribute to the understanding of vapor plume transport in the unsaturated zone and provides valuable experimental data for the transfer to field like conditions.

## 1 Introduction

Subsurface contamination is a major concern in industrialized as well as in developing and emerging countries. NAPL contaminants introduced into the unsaturated zone spread as a liquid phase; however, they can also vaporize and migrate in a gaseous

state. In particular, vapor (gas) plumes migrate easily in the unsaturated zone (Barber and Davis, 1991; Davis et al., 2005, 2009; Höhener et al., 2006). Vapors heavier than air preferentially migrate downward, posing a potential threat to aquifers. When assessing the danger of groundwater contamination by migrating vapor plumes, retention effects on transport are of major interest. Processes such as partitioning to soil water or adsorption on sand grains affect the migration of vapors in the unsaturated





zone. Vapor retardation could potentially slow down migration and reduce the total contaminant mass eventually reaching, and thus endangering the groundwater. While fate and transport of vapor plumes have attracted a great deal of attention over the past years, further (contaminant-specific) investigations are necessary to improve the process understanding required to assess the threat to the environment (Rivett et al., 2011). The contaminant used in this work was carbon disulfide ($CS_2$), an industrial,

non-polar solvent among many used to manufacture viscose rayon. It is highly volatile and characterized by a very high density (1.6 times the air density) in a gaseous state. $CS_2$ has been found in 139 (11.2 %) contaminated sites on the U.S. EPA National Priority List (NPL), according to McGeough et al. (2007).

Experimental studies (e.g. Brusseau et al., 1997; Kim et al., 1998) have been conducted to investigate retardation of the most common VOC in unsaturated porous media. Experimental results have been compared with standard as well as advanced

advection-dispersion models (Popovičová and Brusseau, 1998; Toride et al., 2003). Corley et al. (1996) showed that low concentrations of volatile organic compounds distribute in the bulk phases (air, water and solid), adsorb to the air-water interface, and partition into intraparticle pores in unsaturated and saturated porous media. While it has been demonstrated in experiments that the gas-water interface poses a high potential for retardation (Brusseau et al., 1997), determining the size of interfacial areas and partitioning parameters in theoretical approaches is considered a challenge (Hoff et al., 1993; Kim et al., 1997, 1998).

Mayes et al. (2003) stated that immobile water in pores could act as a short-term sink and as a long-term source of potential contaminants. The effect of moisture content on vapor retention has also been described by Cabbar and Bostanci (2001) and Maxfield et al. (2005) who discovered retardation to be negatively correlated to water saturation due to preferred adsorption on the solid matrix of certain components. The latter has additionally shown the dependency of retardation on the properties of the chemical compound of interest. For instance, noble gases show no retardation behavior at all.

This component and water-saturation-dependent behavior of gas-phase retention emphasizes the necessity for a thorough investigation into retardation of carbon disulfide ($CS_2$) in partially saturated porous media. Thereby, fundamental knowledge regarding its potential to delay or prevent a contamination of an underlying aquifer is gained. Large-scale column experiments were designed and conducted to quantitatively characterize retardation of $CS_2$ with clearly-defined and controlled boundary conditions. The experiments were conducted in vertical columns (i.d. = 0.109 m) of 2 m length packed with a porous medium.

They were carried out under dry conditions and at static water saturations. Reproducible water saturations (initial conditions) were obtained by saturation with water and subsequent drainage under controlled conditions at predetermined capillary pressures. A finite slug containing gaseous $CS_2$ as well as a non-retarding, conservative tracer (argon) was injected via an injection section at the bottom of the column. Effluent concentrations of $CS_2$ and argon were measured online at the top outlet of the column. Tensiometers installed along the column measured capillary pressures to monitor the drainage process and to obtain

water-saturation profiles of the porous medium. Gas flow rates were controlled by mass-flow controllers and additionally measured by a bubble flow meter. This novel experiment set-up enabled for the quantification of $CS_2$ retardation as a function of porous medium, water saturation, and seepage velocity.





**Table 1.** Physicochemical properties of contaminant carbon disulfide ($CS_2$) at $20°C$, $1013.15\,hPa$.

| Parameter | Value | Reference |
|---|---|---|
| CAS number | 75-15-0 | |
| Molecular weight ($M_{CS2}$), $g\,mol^{-1}$ | 76.1 | Budavari (1996) |
| Density of liquid ($\rho$), $kg\,m^{-1}$ | 1263 | Budavari (1996) |
| Solubility in water ($c_{w,sat}$), $g\,L^{-1}$ | 2.1 | Riddick et al. (1986) |
| Henry coefficient ($H_{cc}$), dimensionless | 1.04 | De Bruyn et al. (1995) |
| Boiling point ($T_B$), °C | 46.5 | Budavari (1996) |
| Vapor pressure ($P_{sat}$), hPa | 396.9 | Wagner equation |
| Saturation concentration in gas phase ($c_{a,sat}$), $kg\,m^{-3}$ | 1.239 | Ideal gas law |
| $CS_2$-Air vapor mixture density ($\rho_{vap}$), $kg\,m^{-3}$ | 1.971 | Ideal gas law |
| Diffusion coefficient in air ($D_{CS_2 Air}$), $cm^2\,s^{-1}$ | 9.71e-02 | Chapman-Enskog |

## 2 Materials and methods

This work focused on the experimental investigation into the retardation of $CS_2$ vapor in partially saturated porous media. This sections addresses the experimental set-up, the procedures, and data evaluation methods used in this study. Table 1 shows the physicochemical properties of the contaminant carbon disulfide ($CS_2$) at $20°C$, $1013.15\,hPa$.

### 2.1 Experimental set-up

The experiments were conducted in vertical, stainless steel columns of $2\,m$ length packed with two different types of porous media (Figure 1). The column (length $= 2\,m$, i.d. $= 0.109$m) consisted of two custom-built, $1\,m$ long sections. The ports along the column at a distance of $25\,cm$ allowed for the installation of tensiometers to monitor capillary pressures. At the bottom of the column, the injection section with a base plate was installed. Into this base plate, a porous plate made of recrystallized

silicon carbid was glued to act as a suction plate for the water drainage. The bottom of the column was realized as a constant-mass-flux boundary while the top is open to the surroundings, hence, at constant pressure.

Two different types of porous media (Table 2) were used in the experiments: fine glass beads (soda-lime glass, Sigmund Lindner, Warmensteinach, Germany) and Geba fine sand (Quarzsande GmbH, Eferding, Austria). Their grain-size distributions as well as capillary pressure - water saturation relationships are shown in Figure 2. The columns were packed by dry

pluviation using a sand rainer. The design of the rainer was adopted from Rad and Tumay (1987) with modifications according to Lagioia et al. (2006). The columns consisting of two column sections were filled section after section, each with an overfill of around $30\,cm$. This prevented additional layering of the porous medium. The columns were sealed with cover plates equipped with 1/8" tube fittings (SS-6M0-1-4RT, Swagelok). The experiments were carried out under dry conditions as well as at irreducible, static water saturation. Comparable initial conditions for each experiment were guaranteed by a set-up controlling and

monitoring saturation and drainage. The drainage of the porous media was realized by means of the porous plate at the bottom





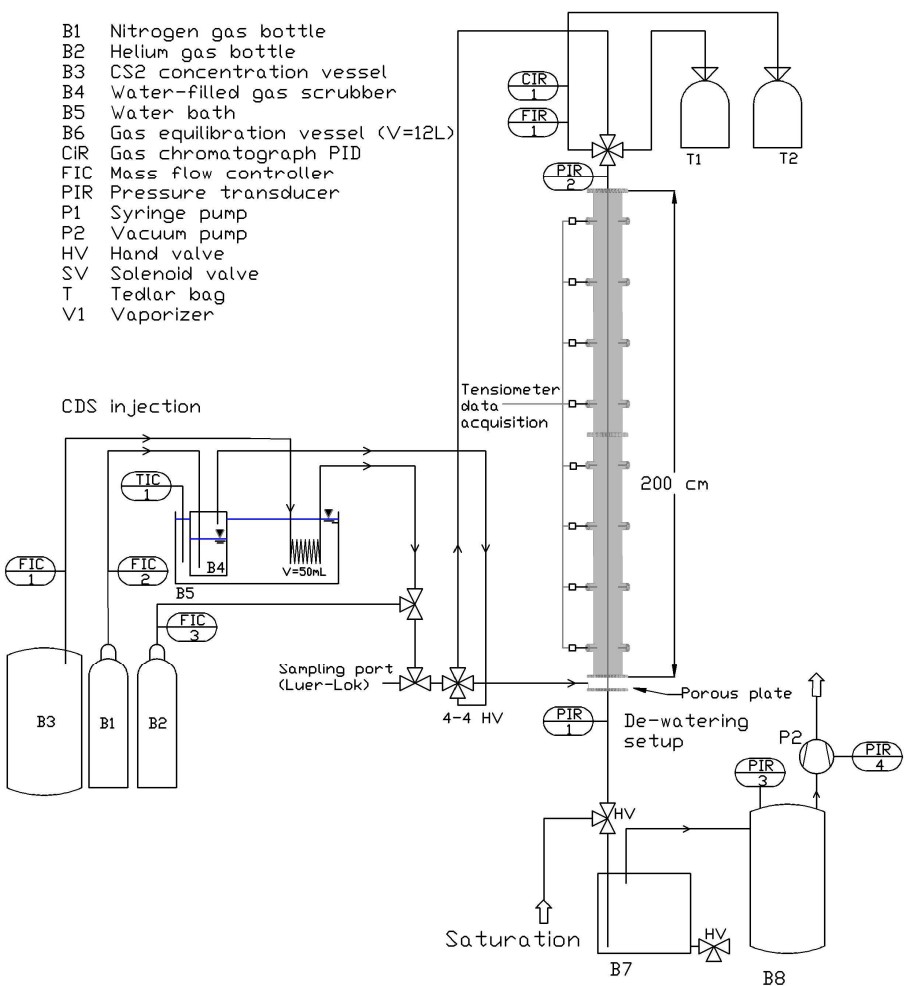

**Figure 1.** Flowchart of vapor-retardation experiment showing column, injection and saturation/drainage set-up.

of the column exclusively permeable for water when fully saturated. This allowed for setting the water Table lower than the bottom of the column or flume. The saturations followed the capillary pressure–saturation relationship (Fig. 2) measured in the laboratory. The water saturation was monitored using the installed tensiometers at the column ports. The tensiometer used for measuring capillary pressures consisted of a ceramic frit (length = 8 mm, o.d. = 6.5 mm, pore size = 2.5 um, porosity = 45 %,

5    Porous Ceramics, Soilmoisture Equipment Corp., Santa Barbara, USA) glued in a stainless steel capillary (length = 200 mm, o.d. = 6 mm, i.d. = 4 mm). In addition, the column was placed on a scale to permanently monitor its weight and thus the total amount of pore water.





**Table 2.** Characteristic properties of the porous media used for the experiments.

| Parameter | Fine glass beads | Geba fine sand |
|---|---|---|
| Bulk density, kg m$^{-3}$ | 1420 | 1390 |
| Grain size, mm | 0.1 – 0.2 | 0.06 – 0.35 |
| Permeability, m$^2$ | $1 \cdot 10^{-11}$ | $1 \cdot 10^{-11}$ |
| Grain diameter $d_{50}$, um | 162 | 140 |
| Pore diameter (median), um | 66 | 39 |
| Knudsen number ($CS_2$) | $1.51 \cdot 10^{-4}$ | $2.56 \cdot 10^{-4}$ |
| van Genuchten (constrained) | | |
| $\alpha$, cm$^{-1}$ | 0.0193 | 0.0145 |
| $n$, dimensionless | 17.783 | 10.305 |
| $\theta_s$, cm$^3$ cm$^{-3}$ | 0.392 | 0.460 |
| $\theta_r$, cm$^3$ cm$^{-3}$ | 0.043 | 0.071 |

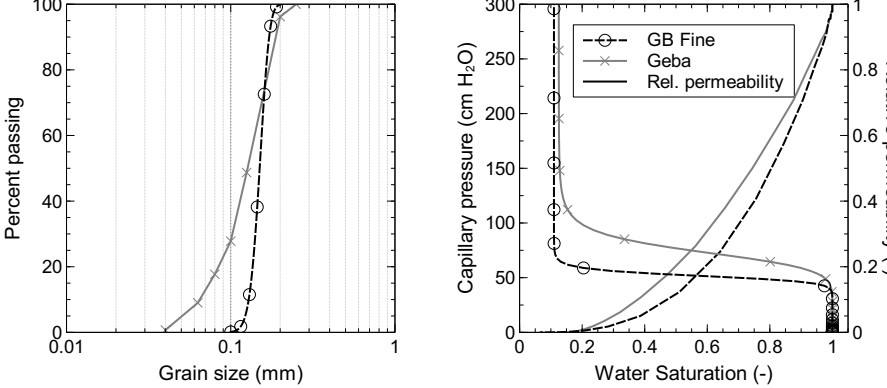

**Figure 2.** Grains-size distribution, capillary pressure–saturation relationship, and relative permeabilities (for the wetting phase) of materials used in experiments.

The injection section at the bottom of the column allowed for the injection of a gas-mixture slug at a predefined mass flux and, in addition, for a controlled upward flow stabilizing the vapor front. The $CS_2$ vapor was prepared prior to injection. A predefined amount of liquid $CS_2$ was injected into a barrel (V = 50 L) and pressurized to an excess pressure of about 2 bar with nitrogen to ensure defined vapor properties. The tracer argon was provided from a gas cylinder (Westfalen AG, Münster, Germany). Constant mass fluxes of the injected $CS_2$ vapor and of the conservative tracer (argon) were critical to the experiment. Mass fluxes of argon, $CS_2$ vapor, and nitrogen were controlled by mass-flow controllers (EL-FLOW, $Q_{max}$ = 3, 50, and 10 mL m$^{-1}$, Bronkhorst High-Tech B.V., Ruurlo, Netherlands). Complete gas tightness of the entire set-up was ensured by using 1/8" stainless steel capillaries throughout. The slug of the gas mixture ($CS_2$, argon, and the carrier nitrogen) was injected



and then pushed through the column using a nitrogen chase at the same flow rate. Prior to injection into the column, argon (approximately $Q_{Ar}/Q_{total} = 1.4\,\%$) was added to the total flow. The mass balance was closed based on the measured flow rate, and the injection and effluent concentrations. The second objective of the nitrogen chase was to observe the recovery of the contaminant and reversibility of partitioning processes. In the case of moist experiments, the gas mixture slug was humidified

with ultra-pure water (RH = 100 %) to avoid a drying-up of the moist porous medium. For the preparation of the gas-mixture slug, a custom-built miniature vaporizer (ICTV, University of Stuttgart, Germany) with an ultra-low volume pump (M6, VICI AG International, Schenkon, Switzerland) was used. The nitrogen used for the chase was bubbled through a gas scrubber filled with ultra-pure water. The inlet steel capillary loop (length = 4 m) and the scrubber were placed in a temperature-controlled water bath (Ministat 125, Huber Kältemaschinenbau GmbH, Germany) to minimize temperature-induced fluctuations during

the experiments.

In the column outflow, $CS_2$ and argon concentration were measured to quantify retardation in dry and moist porous media. Two gas chromatographs were directly connected in-line to the column outlet. $CS_2$ concentrations were determined using a gas chromatograph with a photoionization detector (GC-PID HE1, Meta Messtechnische Systeme GmbH, Dresden, Germany) and argon concentrations were determined using a gas chromatograph with thermal conductivity detector (GC-TCD, Multiple

gas analyzer 8610-0270, SRI Instruments Europe GmbH, Bad Honnef, Germany). Single-point calibrations were conducted prior to and after each run. Measurement intervals were set depending on the flow velocity such that a high temporal resolution (0.021 to 0.065 PV) of the breakthrough curves was obtained. Prior to the start of the slug injection, the concentration of $CS_2$ and argon in the slug mixture was measured as a base to normalize concentrations. A relative pressure transducer connected to the column inlet before the injection section was used to monitor the injection pressure. Since the top column outlet was open

to the atmosphere ($P_{atm}$), this corresponded to the pressure loss caused by the flow through the porous medium. Temperature sensors and absolute pressure transducer continuously measured and recorded ambient and water bath temperature as well as atmospheric pressure in the vicinity of the experiment.

## 2.2  Experimental procedure

Various experiment series were conducted in two different porous media (fine glass beads and Geba fine sand) under both dry

and partially saturated (moist) conditions. Within each series the columns were not repacked and no saturation-and-drainage cycle (SD) was carried out since first tests proved that the partitioning processes were fully reversible. The water saturation or total amount of pore water was monitored throughout the experiment. The slug of the gas mixture was injected with a predefined mass flux into the bottom of the 2 m long column such that it resulted in the designed migration velocity. In each series, experiments were performed with different velocities including 25, 50, 100, and 200 cm h$^{-1}$ (approx. 0.125, 0.25,

0.5, and 1.0 PV h$^{-1}$) to observe effects on retardation by kinetics. A slug of about 3.5 PV was used which corresponded to injection durations of approx. 3.5, 7, 14, and 28 h depending on the respective velocities. This ensured a residence time (plus safety factor) sufficient to attain steady-state conditions and for partitioning processes to reach equilibrium.

The experiments were conducted in four steps. In the first step, the flow rates (slug and chase) were adjusted to match the target migration velocity. In the second step, the column was flushed with nitrogen. While maintaining constant flux, the inflow



was switched to the slug injection of the gas mixture in the third step. After injecting 3.5 PV it was switched back to the nitrogen chase (fourth step).

## 2.3 Data evaluation

The objective of the experiments was to quantify the retardation of $CS_2$ vapor. Possible influences on the determined retardation

factors due to experimental artifacts such as a deviation between theoretical and actual gas-effective pore volume had to be taken into consideration. Hence, for each experiment the breakthrough curve of $CS_2$ was related to that of argon. Concentrations were normalized with respect to the steady-state concentrations ($c = c_{exp}/c_{ss}$). Mass balance was calculated from concentration data and measured gas flow rates. Data was evaluated based on elapsed time and then correlated via flow rate, resulting from mass flux, to gas-effective pore volume. Moreover, both the slug itself and the nitrogen chase were considered which allowed

for an separate evaluation of the slug front and tail (front of nitrogen chase). Breakthrough curves were evaluated using the temporal-moment analysis (TMA) for a step input (slug) as proposed by Yu et al. (1999) and Luo et al. (2006). The advantage of TMA "is that no underlying physical model is needed for calculating the travel times" (Yu et al. (1999), p. 3571), and the breakthrough curves of the $CS_2$-Ar mixture (slug front) as well as the nitrogen chase (or slug tail) can be evaluated individually. Moreover, TMA can also be applied to asymmetrical BTCs resulting from non-equilibrium sorption processes during transport.

Hence, retardation of $CS_2$ in moist and dry porous media could be compared and the impact of water saturation on retardation of $CS_2$ could be delineated.

TMA was applied to obtain transport parameters (seepage velocity [Eq. A9] and dispersion coefficient [Eq. A10]) and mean breakthrough arrival (Eq. A7) time from concentration breakthrough curves. The retardation factor $R$ of $CS_2$ vapor was calculated from the ratio of the respective moments or mean breakthrough arrival time.

$$R = \frac{\tau_{CS_2}}{\tau_{Ar}} = \frac{M_{1,CS_2}}{M_{1,Ar}} \tag{1}$$

This ensured the independence from experimentally-induced deviations and thus allowed to quantify the influence of water saturation and migration velocity on retardation. Experimental retardation factors were compared with a theoretical factor. Brusseau et al. (1997) used carbon dioxide ($CO_2$) as a tracer whose predominant source of retardation was the partitioning into the aqueous phase. The similarity of $CS_2$ and $CO_2$ regarding solubility in water and Henry constant suggests a comparable

retardation behavior for $CS_2$. Hence, partitioning into the aqueous phase is considered the only contribution. Adsorption on grains (3) and at the gas-water interface (4) (terms on the right hand-side of Eq. A11) will be neglected. This then yields the adapted theoretical retardation coefficient.

$$R_t = 1 + \frac{\theta_w}{\theta_a K_H} \tag{2}$$

where $\theta_w$ is volumetric water content, $\theta_a$ is gas-filled porosity, $K_H$ (dimensionless) is Henry's constant.





**Table 3.** Experimental conditions of vapor-retardation experiments in fine glass beads and Geba fine sand in dry and moist conditions (series).

| Series | 1 | 2 | 3 | 4 |
|---|---|---|---|---|
| Condition | dry | moist | moist | moist |
| | Fine glass beads | | | |
| Porosity ($\phi$) | 0.40 | 0.40 | 0.40 | 0.40 |
| Mean water saturation ($S_w$) | 0.0 | 0.088 | 0.154 | 0.073 |
| Eff. pore volume, L | 7.72 | 7.04 | 6.53 | 7.16 |
| | Geba fine sand | | | |
| Porosity ($\phi$) | 0.40 | 0.40 | 0.40 | - |
| Mean water saturation ($S_w$) | 0.0 | 0.162 | 0.150 | - |
| Eff. pore volume, L | 7.58 | 6.35 | 6.45 | - |

## 3 Results and discussion

Column experiments were conducted with dry and moist porous media to characterize retardation of $CS_2$ vapor. Table 3 shows the experimental conditions of each series performed in fine glass beads and Geba fine sand. Several series of experiments were performed in each porous medium to quantify retardation. Series 1 refers to the experiments conducted in dry porous media while Series 2 to 4 refer to the experiments in moist conditions. A saturation-and-drainage cycle was performed prior to each moist series. A slug of 3.5 PV of the gas mixture was injected ensuring, even for high flow rates, a sufficient residence time to reach equilibrium in the 2 m long column. Different seepage velocities (25, 50, 100, and $200 \, \mathrm{cm \, h^{-1}}$) were applied, based on previously conducted experiments investigating density-driven vapor migration (Kleinknecht et al., 2015). Breakthrough curves under the prevailing experimental conditions were determined from concentration measurements at the column outlet. The temporal-moment analysis (Sec. 2.3) was applied to the breakthrough curves (BTC) to quantify diffusion/dispersion and retardation as a function of the porous media, the water saturation, and the flow conditions. A detailed summary of all experiments (experimental conditions, injected mass, and mass recovery) is given in Tables 5 and 6 in the appendix.

### 3.1 Water saturations

The moist porous medium required for this investigation was obtained by saturation and subsequent drainage. The suction applied at the bottom of the column during drainage was responsible for the observed water saturation profiles. The capillary pressure was measured with the tensiometers installed at the column ports to derive water saturations along the column ($P_c$–$S_w$, Fig. 2). Mean water saturations of the moist series are given in Table 3.

Figure 3 (left-hand) shows the initial water saturation profiles along the column measured in fine glass beads (only Series 4) and Geba fine sand (Series 2 and 3). Unfortunately, no tensiometer measurement data was available for Series 2 and 3 in fine glass beads. However, the available profile of Series 4 revealed a uniform saturation along the column, only slightly increasing





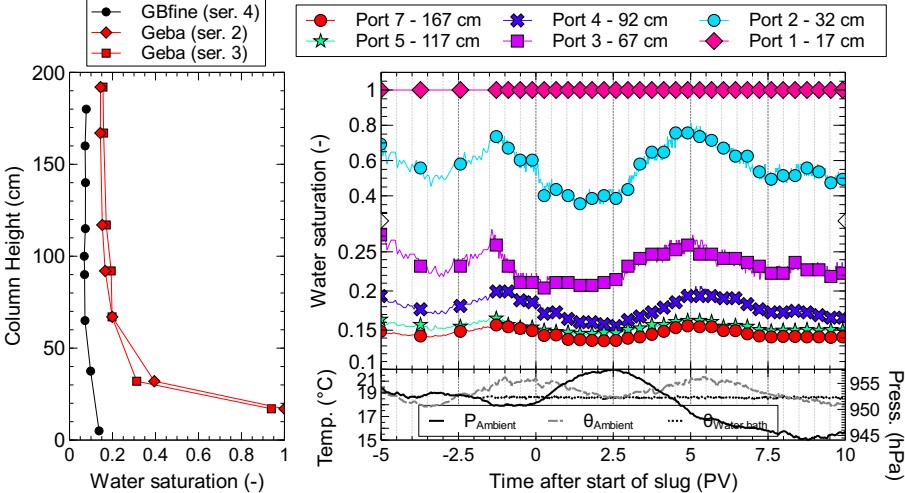

**Figure 3.** Initial water saturation and progression of water saturation in experiment with Geba fine sand (Series 2).

toward the bottom from $S_w = 0.07$ to $0.14$. The very narrow and uniform grain-size distribution of the fine glass beads was responsible for a sharp transition from full to irreducible saturation (see $P_c$–$S_w$ curve in Fig. 2), thus favoring a uniform saturation profile. In Geba fine sand, a constant water saturation of $S_w = 0.15$ above a column height of $70\,cm$ was measured. However, both profiles showed a pronounced increase in the water saturation toward the bottom of the column, apparently

reaching fully-saturated conditions according to the $P_c$–$S_w$ relationship of Geba fine sand (Fig. 2). Still, capillary pressures of $P_c = 55$ and $65\,hPa$ were measured at the lowest port. The suction applied via the porous plate was limited by its air entry pressure. A further decrease of pressure would have resulted in a breakthrough of air (continuous gas phase). The mean water saturation (Table 3) but also the observed profiles were expected to have an impact on the retardation behavior of $CS_2$, as discussed in Sec. 3.3.

Figure 3 (right-hand) shows an exemplary progression of water saturations measured in Experiment 28 of Series 2 with Geba fine sand (Table 6). The saturations were based on tensiometers along the column during the injection of the slug and the subsequent nitrogen chase. The tensiometers suggested an apparent change in water saturation during active gas flow through the porous medium. This was most likely provoked by the pressure increase due to the injection and the gas flow around the tensiometer. It is important to note that the tensiometers measured the suction at a very spatially-limited location in the

porous medium due to the small size of their tips (o.d. $= 6\,mm$, length $= 8\,mm$). In addition, the pressure transducers of the tensiometers showed periodic fluctuations as a result of daily temperature changes in the laboratory hall and due to varying ambient pressure. However, a drying-out of the porous medium was prevented by the humidification of all gases (slug and chase) prior to injection. This was confirmed by the water mass balance by means of the continuous weight measurement of the entire column throughout all experiments conducted within a series. Hence, the initial water-saturation profile could





## 3.2 Impact of velocity on breakthrough

The impact of the seepage velocity on the concentration breakthroughs of argon and CS$_2$ was investigated. Thus, different
velocities were applied to characterize the transport. Figure 4 and 5 show breakthrough curves of CS$_2$ and argon as a function
of pore volume for different flow conditions (velocities) in moist fine glass beads and Geba fine sand. The breakthrough curves
were adapted to the actual gas-effective pore volume determined from the mean breakthrough arrival time $\tau_{Ar}$ (Eq. A7) of the
conservative tracer argon. Seepage velocities of about 25, 50, 100, and 200 cm h$^{-1}$ (residence time of about 8, 4, 2, and 1 h)
were applied successively in the same column and under similar initial conditions. The lines represent measured concentrations
$(c/c_{ss})$ normalized to steady-state concentration. The graphs are split and the right-hand side shows the outflow concentrations
after the injection was switched from the gas-mixture slug to the nitrogen chase. Thus, these experiments allowed for the
individual evaluation of the slug and of the nitrogen chase breakthroughs. The skewness of a BTC is a result of the longitudinal
molecular diffusion and the mechanical mixing. The BTCs of argon and CS$_2$ shown in the graphs illustrate that the skewness
increased with decreasing seepage velocity as a result of increased diffusion during the longer residence time. Since argon
was used as a conservative tracer, its breakthrough was a function of the seepage velocity only. CS$_2$ was additionally affected
by retardation, hence its breakthrough depended on seepage velocity as well as water saturation. The retardation of CS$_2$ is
discussed in detail in the following Section 3.3. The repetitions with a velocity of about 50 cm h$^{-1}$ proved that equilibrium
was reached and they showed good reproducibility of the experiments.

The BTCs were evaluated with the temporal-moment analysis (TMA, see Sec. A3) to obtain dispersion coefficients (Eq. A10)
of argon and CS$_2$ for different flow conditions. Figure 6 shows dispersion coefficients as a function of velocity of these ex-
periments in moist porous media. Dispersion coefficients of argon and CS$_2$ increased from $D_{Ar} = 0.089$ to $0.142$ cm$^2$ s$^{-1}$ and
$D_{CS_2} = 0.033$ to $0.074$ cm$^2$ s$^{-1}$ as a function of the seepage velocity and the porous medium. The dispersion coefficient is de-
fined as $D = D^* + \alpha v$ (Eq. A5). Under static conditions (v = 0 cm h$^{-1}$), the effective binary diffusion coefficients $D^*$ in porous
media should apply. $D^*$ is defined as the product of the binary diffusion coefficient (Eq. A1) of the component in nitrogen and a
tortuosity factor (Eq. A4). In the case of flow, the dispersion coefficient increases due to mechanical mixing which is a measure
of the heterogeneity of the porous medium or the flow region, respectively. It is defined as the product of the dispersivity $\alpha$ and
the velocity.

The effective binary diffusion coefficients $D^*$ and the dispersivity $\alpha$ were determined from the breakthrough curves of
the experiments. Based on the equation above, a linear regression was fitted to the dispersion coefficients as a function of
the velocity for each porous medium. The y-intercepts of the regression lines represent the coefficient $D^*$ and the slopes
express dispersivity $\alpha$ of the respective porous medium. The theoretical coefficient $D_t^*$ was determined according to the
Chapman-Enskog theory and the approach by Millington and Quirk (1961) which accounts for tortuosity due to porous ma-
trix and water saturation (see Sec. A1). Table 4 compares theoretical with experimental effective binary diffusion coefficients
of CS$_2$ and argon in fine glass beads and Geba fine sand under the given experimental conditions (water saturation S$_w$ and





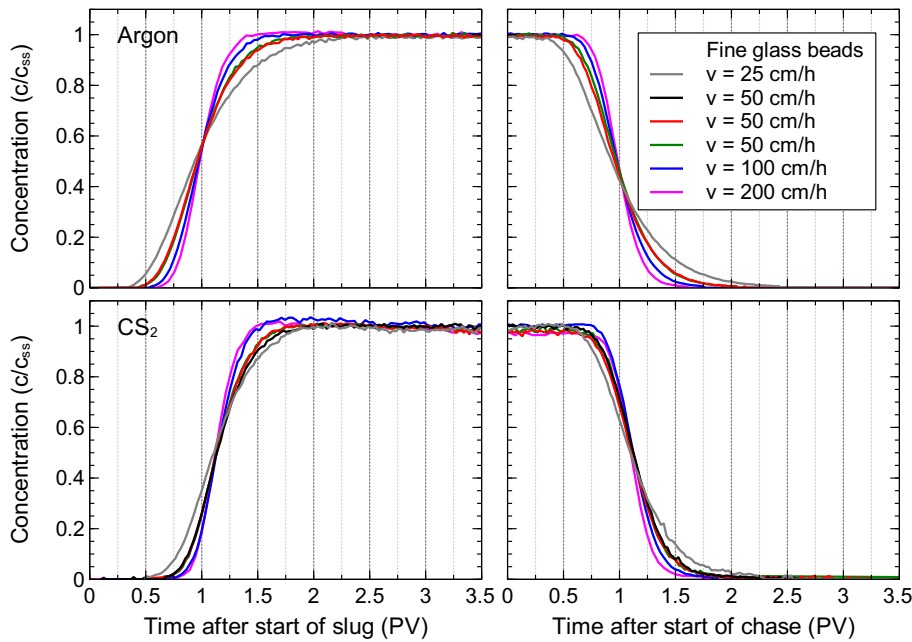

**Figure 4.** Breakthrough curves of $CS_2$ and Ar in moist fine glass beads ($S_w = 0.088$) for different velocities.

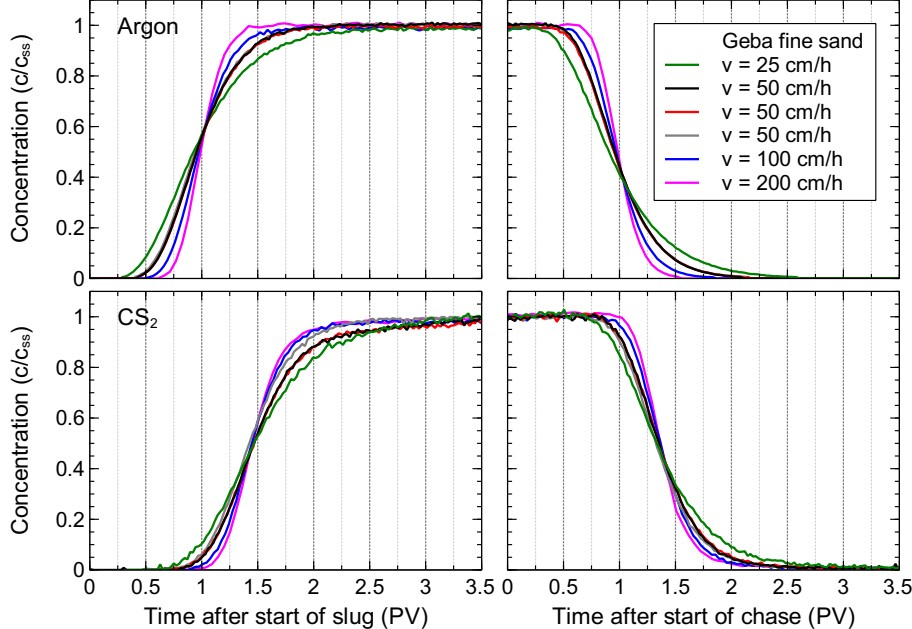

**Figure 5.** Breakthrough curves of $CS_2$ and Ar in moist Geba fine sand ($S_w = 0.15$) for different velocities.





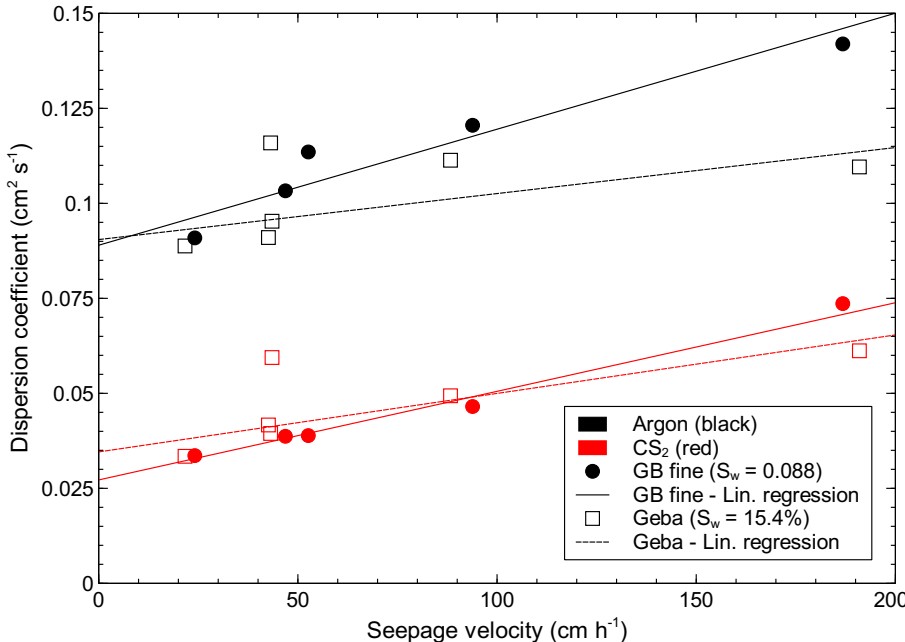

**Figure 6.** Dispersion coefficients of $CS_2$ and Ar determined from TMA as a function of velocity. Experiments were conducted in fine glass beads ($S_w = 0.088$, Series 2) and Geba fine sand ($S_w = 0.154$, Series 2).

**Table 4.** Theoretical and experimental effective binary diffusion coefficient $D^*$ of argon and $CS_2$, dispersivity $\alpha$, and coefficient of determination $R^2$ of linear regression determined from experiments in moist porous media (Series 2).

| Porous medium | | Fine glass beads | Geba fine sand |
|---|---|---|---|
| Water saturation $S_w$ | | 0.088 | 0.154 |
| Tortuosity $\tau$ | | 0.220 | 0.161 |
| Argon | $D_t^*$, cm$^2$ s$^{-1}$ | 0.0386 | 0.0284 |
| | $D^*$, cm$^2$ s$^{-1}$ | 0.0909 | 0.0966 |
| | $\alpha$, cm | 1.029 | 0.313 |
| | $R^2$ (lin. regression) | 0.919 | 0.254 |
| $CS_2$ | $D_t^*$, cm$^2$ s$^{-1}$ | 0.0213 | 0.0157 |
| | $D^*$, cm$^2$ s$^{-1}$ | 0.0263 | 0.0332 |
| | $\alpha$, cm | 0.888 | 0.552 |
| | $R^2$ (lin. regression) | 0.987 | 0.952 |

tortuosity $tau$). In fine glass beads, effective binary diffusion coefficients of argon were $D_{Ar}^* = 0.0909\,\mathrm{cm^2\,s^{-1}}$ compared to $D_{t,Ar}^* = 0.0386\,\mathrm{cm^2\,s^{-1}}$ and of $CS_2$ were $D_{CS_2}^* = 0.0263\,\mathrm{cm^2\,s^{-1}}$ compared to $D_{t,CS_2}^* = 0.0213\,\mathrm{cm^2\,s^{-1}}$. In Geba fine sand, co-





efficients of argon were $D^*_{Ar} = 0.0966 \, \text{cm}^2 \, \text{s}^{-1}$ compared to $D^*_{t,Ar} = 0.0284 \, \text{cm}^2 \, \text{s}^{-1}$ and of $CS_2$ were $D^*_{CS_2} = 0.0332 \, \text{cm}^2 \, \text{s}^{-1}$ compared to $D^*_{t,CS_2} = 0.0157 \, \text{cm}^2 \, \text{s}^{-1}$. The experimental coefficients $D^*$ differed from the theoretical effective binary diffusion coefficient $D^*_t$ calculated for the prevailing conditions. This could result from the choice of porous media, since both media were characterized by a uniform and narrow grain-size distribution, as well as the observed water-saturation profiles.

Werner et al. (2004) reported that theoretical approaches are often sensitive since the majority of their parameters are raised to a high power and do not apply satisfactorily to a wide variety of soils. Furthermore, the theoretical approach does not take into account material characteristics such as the pore-size distribution which may vary for similar porosities and hence affect the tortuosity factor. Dispersion coefficients shown in Figure 6 varied for a given velocity due to minor differences between the experiments and to variations arising from the temporal-moment analysis. The equation used to determine the dispersion

coefficient (Eq. A10) from TMA raises the velocity to the power of three, thus minor deviations had a great impact on the final values.

  The increase in the dispersion coefficient in Figure 6 from the effective binary diffusion coefficient (at $v = 0 \, \text{cm} \, \text{h}^{-1}$) with increasing velocity resulted from mechanical mixing due to flow through the moist porous medium. This was observed in all experiments. The increase is determined by the slope of the linear regression representing the dispersivity $\alpha$ which should be

a parameter of the porous medium only and should be independent of the components (gases) and flow conditions. A slight difference was found between $CS_2$ and argon for both materials, resulting in a mean dispersivity of $\alpha_{\text{GBfine}} = 0.958 \, \text{cm}$ in fine glass beads and $\alpha_{\text{Geba}} = 0.432 \, \text{cm}$ in Geba fine sand. The difference could be due to dispersivity transforming from a physical system to a lumped parameter, because of e.g. diffusional or nonequilibrium effects. This then results in a component-dependent dispersivity according to Costanza-Robinson and Brusseau (2002), who reported that dispersivity ranges from approx. 0.1

to 5.0 cm. Since argon is a conservative tracer and $CS_2$ is affected by retardation, greater reliability was attributed to the dispersivity $\alpha_{Ar}$ determined from BTCs of argon. The results of the experiments in this work demonstrate the impact of seepage velocities on the diffusion/dispersion of $CS_2$ vapor and of argon. Thus, an influence of the velocity on the retardation of $CS_2$ was expected.

### 3.3 Retardation of $CS_2$

Different series of experiments were conducted to quantify retardation of $CS_2$ as a function of water saturation and seepage velocity. Figure 7 and 8 compare breakthrough curves (BTC) of argon and $CS_2$ in dry (black) and moist (red) porous medium for the same seepage velocity ($v = 50 \, \text{cm} \, \text{h}^{-1}$). Two to three repetitions of each run in dry and moist conditions, respectively, are shown. The graphs show normalized concentrations as a function of effective pore volume (total pore volume minus water content after drainage).

The BTCs of argon showed excellent reproducibility in repetition experiments in both materials at the same conditions ($v = 50 \, \text{cm} \, \text{h}^{-1}$). In fine glass beads, argon showed very similar BTCs for dry and moist experiments. This thus confirmed that argon experiences no retardation and may be used as a conservative tracer and as a reference for $CS_2$. In Geba fine sand, a different skewness was observed between dry and moist conditions as a result of the reduced pore space in moist conditions. Hence, a comparison of the BTCs revealed that the effective-flow region in fine glass beads was similar in dry and moist



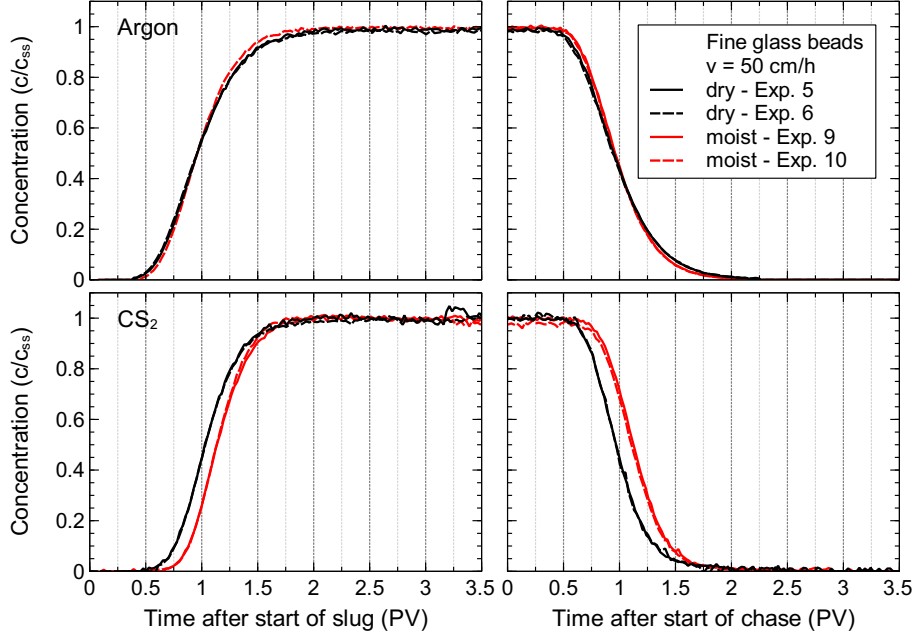

**Figure 7.** Breakthrough curves of CS$_2$ and Ar in dry and moist (S$_w$ = 0.088) fine glass beads under identical slug and flow conditions (v = 50 cm h$^{-1}$).

conditions, whereas in Geba fine sand it was reduced in moist conditions. This resulted in BTCs which were less affected by diffusion due to a shorter residence time. Since the experiments were conducted with a constant-flow-rate boundary condition based on the calculated effective pore volume, a shorter residence time i.e. higher seepage velocity occurred when the actual effective pore volume available for gas flow is smaller than the calculated volume.

The BTCs of CS$_2$ showed, in general, good reproducibility for all experiments. In fine glass beads, a later breakthrough of CS$_2$ compared to argon can be observed in Figure 7, demonstrating the retardation of CS$_2$ due to partitioning into the water phase. The different effective pore volume due to the pore water and possible reduced residence time (actual vs. calculated PV) resulted in less skewed BTCs compared to the dry experiments. In Geba fine sand, a more pronounced retardation of CS$_2$ was observed compared to experiments in fine glass beads. The later breakthrough becomes evident when comparing BTCs

in dry (black) with moist (red) conditions in Figure 8. This could be ascribed to the overall higher water saturation and the increase in saturation toward the bottom of the column (see Fig. 3). In two of the three BTCs in moist experiments (Fig. 8), CS$_2$ concentrations leveled at around c/c$_{ss}$ = 0.9 followed by an increase to steady-state (plateau) concentrations toward the end of the slug. This behavior might be a consequence of the water saturation over column height affecting the partitioning processes.

   The retardation coefficients of CS$_2$ as a function of porous medium, water saturation, and seepage velocity were determined

using the temporal-moment analysis (TMA) of the breakthrough curves (see Sec. 2.3). The coefficients were normalized with respect to the BTCs from dry porous medium. Thereby, errors due to set-up or other systematic errors could be eliminated





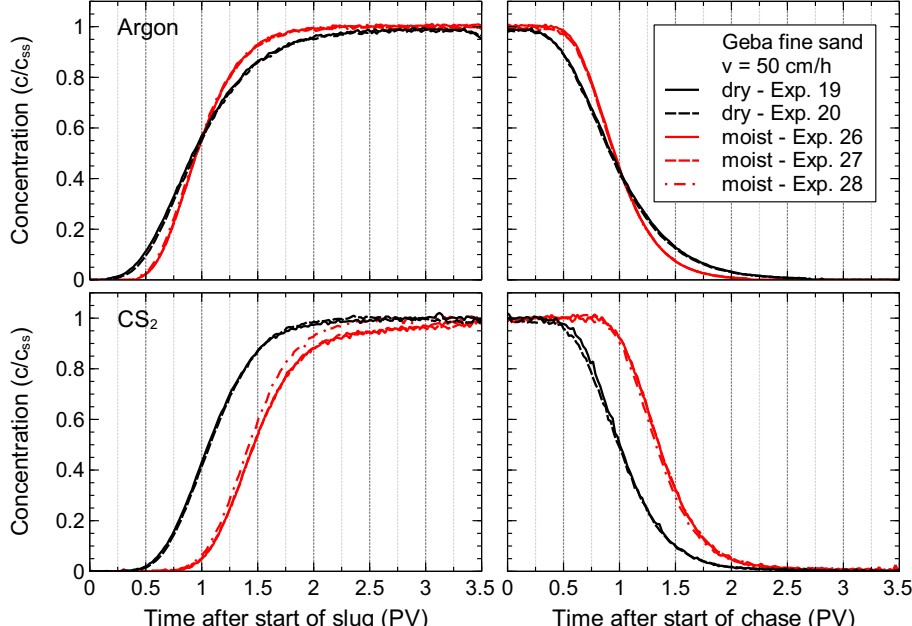

**Figure 8.** Breakthrough curves of $CS_2$ and Ar in dry and moist ($S_w = 0.154$) Geba fine sand under identical slug and flow conditions ($v = 50\,\mathrm{cm\,h^{-1}}$).

and allowed for the comparison with theoretical values. Figure 9 shows retardation coefficients of $CS_2$ as a function of water saturation (upper) and seepage velocity (lower) in fine glass beads (black) and Geba fine sand (red). The coefficients of the slug (circle) and the chase (rectangle) are given and their size represents seepage velocity or water saturation. Note the broken x-axis (water saturation) between $S_w = 0.10$ and $0.13$ in the upper graph indicated by the vertical, dashed lines.

In fine glass beads, a non-linear increase in the retardation coefficient from $R_{GBfine} = 1.09$ to $1.16$ with increasing water saturation from $S_w = 0.075$ to $0.155$ was observed. Of course, partitioning to the water phase is dependent on the gas-water interfacial area which should decrease with increasing water saturation. Thus an extrapolation of the retardation coefficient to higher water saturations might be difficult. The retardation of the slug and of the chase were different in fine glass beads, the chase being more prone to retardation than the slug. The breakthrough of the nitrogen chase (removal of the $CS_2$ vapor)

showed a higher retardation by a factor (average) of $1.05$ compared to the breakthrough of the slug throughout all experiments in fine glass beads. This behavior can be also seen when comparing the BTCs of $CS_2$ in the upper graph of Figure 7.

In Geba fine sand, higher retardation coefficients compared to fine glass beads were measured in the experiments. These ranged between $R_{Geba} = 1.29$ and $1.34$ at a mean water saturation of $S_w = 0.162$. This was due to the higher water saturation and its increase toward the bottom (discussed in Sec. 3.1), the different gas-water interfacial area, and the pore space available for

gas flow. Series 3 in Geba fine sand had to be excluded from these graphs due to mass balance issues discussed later. Hence,





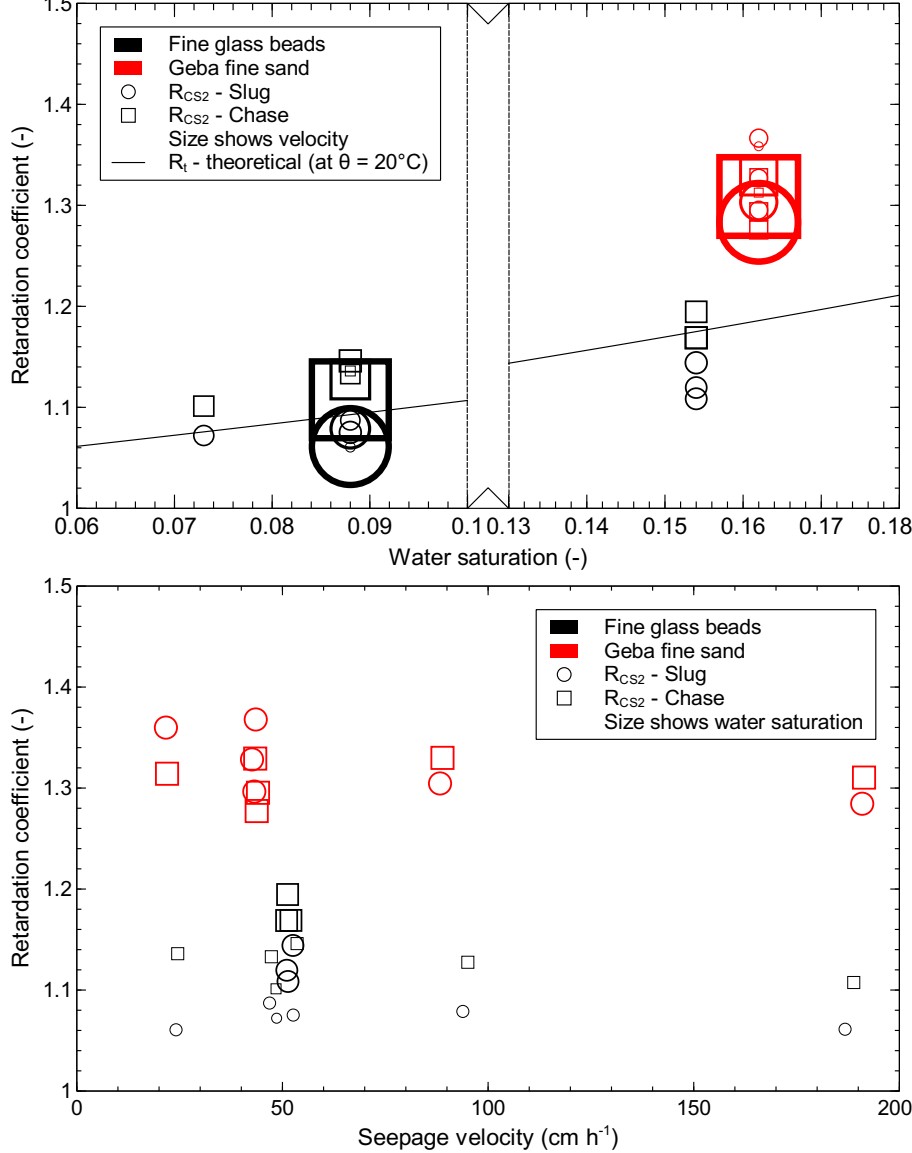

**Figure 9.** Retardation coefficients of $CS_2$ determined from experiments with different seepage velocities in fine glass beads and in Geba fine sand at different water saturations (evaluated with temporal-moment analysis).

results were only available for one particular water saturation in Geba fine sand. The ratio between the retardation coefficient of the slug and that of the chase did not show a clear trend as observed in fine glass beads despite the differences seen in Figure 9.

The retardation coefficients gained from the experiments were compared to a theoretical approach. The adapted theoretical retardation coefficient (Eq. 2) is shown as a line in the upper graph. The theoretical coefficient is calculated taking into account

5  the porosity of the porous medium, the water saturation, and the Henry coefficient. Hence, only one function is shown in the





upper graph of Figure 9, since the porosities of the fine glass beads and the Geba fine sand used were similar. In fine glass beads, the theoretical coefficient compared very well with the values from the experiments. It slightly overpredicted the retardation of the slug while it underpredicted that of the chase, however it reproduced satisfactorily the mean retardation coefficient and its increase with water saturation. In Geba fine sand, the theoretical coefficient significantly underestimated the observed

retardation. This could be due to the fairly simple theoretical approach only taking into account the porosity of the porous medium. It is obvious that the pore-size distribution depending on the grain-size distribution of the porous medium determines the gas-water interfacial area and thus has a significant impact on retardation. However, such material characteristics are not factored in by the theoretical coefficient. Moreover, the non-uniform water-saturation profile along the column height could be responsible for varying partitioning. Finally, deviations could occur due to adsorption processes or higher-order kinetics during

partitioning which were neglected in the theoretical factor. These findings suggest that retardation may vary along the depth of the unsaturated zone due to spatially-varying water saturations and especially around the capillary fringe in the vicinity of the groundwater level.

The experiments were conducted with different seepage velocities to evaluate their impact on retardation. The lower graph in Figure 9 shows the retardation coefficients as a function of seepage velocity. A mean retardation coefficient of $R_{GBfine} =$

$1.100 \pm 0.0096$ and $R_{Geba} = 1.315 \pm 0.0152$ was measured in the experiments with fine glass beads ($S_w = 0.088$) and Geba fine sand ($S_w = 0.162$), respectively. In general, no significant change of the retardation behavior with increasing seepage velocity was observed. This confirmed that the mass transport rate was low enough and the residence time of the slug was sufficient for the partitioning processes to reach equilibrium. Concluding from the experiment with $v = 200 \, \mathrm{cm \, h^{-1}}$ it seems likely that there was a slight tendency toward a reduced retardation. In fact, retardation may reduce at higher seepage velocities due to limiting

contaminant diffusion. If no equilibrium is reached in case of high velocities, the retardation coefficient reflects an apparent coefficient since in this case it is a function of the experimental system used (i.e. length of the column). Additional experimen-tal repetitions would have been required to provide proof. However, this experimental investigation aimed at characterizing retardation of $CS_2$ in the range of seepage velocities observed during vapor-plume migration experiments ($v \ll 200 \, \mathrm{cm \, h^{-1}}$). Hence, the focus laid on the velocities used and higher values were beyond the scope.

Mass balance analyses were performed to obtain mass recovery (r) from each breakthrough curve. Mass recovery was calculated from concentration and flow measurements and were normalized with respect to the injected mass. In general, mass recoveries of argon and $CS_2$ showed good results. The mean recovery of argon calculated from all vapor-retardation experiments conducted yielded $r_{Ar} = 0.995 \pm 0.007$ and confirmed complete mass recovery. The mean recovery of $CS_2$ was $r_{CS_2} = 0.981 \pm 0.084$ (without the experiments of Series 3), thus only suggesting slight mass losses. Mass recoveries of all ex-

periments are given in Tables 5 and 6 in the appendix. The mass balance and complete recovery proved the reliability and quality of the results gained from these column experiments.

The results discussed above excluded Series 3 conducted in Geba fine sand. Series 3 was the second saturation and drainage cycle which was carried out to establish a different static water saturation than in Series 2. However, significant $CS_2$ mass losses became more pronounced with each experiment in this series, eventually leading to its exclusion from the results.



Recoveries of $CS_2$ decreased from $r_{CS_2} = 0.854$ in the first experiment of Series 3 down to $r_{CS_2} = 0.010$. This mass loss of $CS_2$ was caused by biodegradation which was confirmed by the smell of hydrogen sulfide in the column outflow. Cox et al. (2013) found carbonyl sulfide (COS) and hydrogen sulfide ($H_2S$) as by-products during $CS_2$ biodegradation in their experiments. The mass balance analysis of the experiments enabled for determining mean degradation rates of $CS_2$ which were calculated from the $CS_2$ mass rate and the recovery. The mean degradation rates ranged from 0.12 to $1.28\,\mathrm{mg\,h^{-1}}$ depending on the respective seepage velocity applied in the experiments. The experiments showed that biodegradation may have a considerable potential for mitigating the contaminant mass transfer by vapor migration to the underlying aquifer. However, the quantification of biodegradation of $CS_2$ was beyond the scope of this work but should be addressed in future research.

## 4  Conclusions

– The retardation of $CS_2$ vapor was quantified in 2 m long column experiments as a function of porous medium, water saturation, and seepage velocity. The novel set-up and methods applied additionally allowed for characterizing the transport of $CS_2$ and argon.

– The versatile temporal-moment analysis (TMA) was successfully applied to quantify diffusion/dispersion of $CS_2$ and argon as well as retardation of $CS_2$ from concentration breakthrough curves by comparison with the conservative tracer argon.

– Dispersion coefficients as a function of seepage velocity were obtained from the TMA for experiments in moist conditions. Linear regressions of these data sets yielded effective binary dispersion coefficients and dispersivity values at the prevailing experimental conditions. The effective binary diffusion coefficient at the given experimental conditions was found to be slightly higher than theoretical values based on the approach by Millington and Quirk (1961). The theoretical approach takes into account the porosity only and neglects material characteristics such as grain-size or pore-size distribution which affect diffusion/dispersion. Thus, the experiments confirm that theoretical approaches do not satisfactorily apply to a wide variety of porous media (Werner et al., 2004).

– The impact of different seepage velocities on the breakthrough curves and thus on the dispersion coefficient was observed. The experiments showed that the velocities affected diffusion/dispersion of the gases due to the corresponding residence time in the porous medium and due to mechanical mixing. This effect was illustrated by the skewness of the breakthrough curves which were negatively correlated to the seepage velocity.

– The retardation coefficient of $CS_2$ increased with increasing water saturation and compared very well with the theoretical approach for fine glass beads. A slightly higher retardation of the chase by a factor of 1.05 compared to that of the slug was observed. A pronounced higher retardation was observed in Geba fine sand due to the different grain-size distribution and the particular water-saturation profile. Moreover, the theoretical coefficient underpredicted retardation in Geba fine sand. Retardation coefficients as a function of (seepage) velocity revealed only a minor dependency and suggested a slight tendency toward a reduced retardation at higher velocities.





- – Clear evidence of the biodegradation of $CS_2$ was found in the last series of experiments in Geba fine sand confirmed by the concentration measurements and the mass balance analysis. These findings demonstrate the potential of biodegradation to reduce the total $CS_2$ mass in case of a contamination in the unsaturated zone and of migrating vapor plumes eventually threating the underlying aquifer.

- – The experiments conducted clearly proved that a migrating $CS_2$-vapor plume in the unsaturated zone is retarded and that dissolved $CS_2$ is amenable to biodegradation. The breakthrough of the slug and of the chase was observed and evaluated, the latter demonstrating a complete removal of the gaseous $CS_2$ confirmed by mass balance analyses. This observation clearly promotes the remediation of a liquid $CS_2$ spill in the unsaturated zone using soil-vapor extraction.

**Data availability**

The experimental data used to produce the results and graphs presented in this manuscript is available at http://dx.doi.org/10.4228/ZALF.2013.295 (Kleinknecht, 2016).

**Appendix A:  Materials and methods**

**A1  Binary diffusion coefficient**

The Chapman-Enskog formula is used to estimate the binary diffusion coefficient of component $A$ in $B$ at low density.

$$D_{AB} = 1.8583 \times 10^{-3} \frac{\sqrt{T^3 \left( \frac{1}{M_A} + \frac{1}{M_B} \right)}}{p\sigma_{AB}^2 \Omega_{D,AB}} \tag{A1}$$

with $D_{AB}$ (cm$^2$ s$^{-1}$), temperature $T$ (K), pressure $p$ (atm), the Lennard-Jones parameter $\sigma_{AB}$ (angstrom), and the collision integral $\Omega_{D,AB}$ which can be approximated with the Lennard-Jones potential. Component-specific values to determine $\sigma_{AB}$ as well as $\Omega_{D,AB}$ as a function of $kT/\epsilon$ can be found in Bird et al. (1960).

Porous media affect diffusion of gases since space is occupied by grains and possibly by additional fluid phases. Therefore,
Fick's law is often modified by the factor $\beta$ to account for these deviations.

$$D^* = \beta D_{AB} \tag{A2}$$

while $\beta$ is defined as

$$\beta = \phi S_g \tau \tag{A3}$$

where $D_{AB}^*$ is the effective diffusion coefficient in porous media, $\phi$ is the porosity, $S_g$ the gas saturation (equal to 1 for all-gas
condition), and $\tau$ is the tortuosity. According to Millington and Quirk (1961), tortuosity can be approximated by

$$\tau = \phi^{1/3} S_g^{7/3} \tag{A4}$$





## A2 Dispersion coefficient

Flow of fluids in a porous medium may vary significantly on a micro scale due to the velocity field in pores, irregularities of the pore size, flow restrictions, or dead-end pores resulting in additional spreading denoted as dispersion. These influences have to be taken into account in analytical or numerical solutions of flow in porous media. This is done by introducing the longitudinal dispersion coefficient

$$D = \beta D_{AB} + \alpha v = D^* + \alpha v \tag{A5}$$

with the dispersion coefficient $D$ (cm$^2$ s$^{-1}$), effective binary diffusion coefficient $D^*_{AB}$ (cm$^2$ s$^{-1}$) according to Eq. A2, gas-phase longitudinal dispersivity $\alpha$ (cm), and average gas velocity $v$ (cm s$^{-1}$).

## A3 Temporal-moment analysis

The measured breakthrough curve (BTC) data had to be prepared to allow for the usage of the temporal-moment analysis (TMA) generally applied to responses from dirac input. The breakthrough curves of the step-input boundary condition (1) were transformed to a dirac-input boundary condition (2) (Yu et al., 1999). This was achieved by using the derivative of the original step-input BTC data.

$$\frac{\partial c_1}{\partial t} = c_2 \tag{A6}$$

This transformation then allowed for analyzing the original breakthrough curves and required adapted definitions of the temporal moments. The first order normalized moment $M_1$ representing the mean breakthrough arrival time ($\tau$) is then defined as

$$\tau = M_1 = \frac{m_1}{m_0} = \frac{\int_0^1 t\, dc_1}{\int_0^1 dc_1}, \tag{A7}$$

where $c_1$ (-) is normalized concentration of measured BTC and $t$ (second or PV) is elapsed time. The second central moment $\mu_2$ corresponds to the variance of travel times at the location of measurement and is given by

$$\mu_2 = \int_0^1 (t - M_1)^2\, dc_1. \tag{A8}$$

These two moments can be used to directly infer seepage migration velocity $v$ and dispersion coefficient $D$ from BTC data for a one-dimensional system (Cirpka and Kitanidis, 2000).

$$v = \frac{z}{M_1} \tag{A9}$$

$$D = \frac{\mu_2 v^3}{2z} \tag{A10}$$



## A4 Theoretical retardation coefficient

Brusseau et al. (1997) defined the retardation factor to be the sum of various processes including adsorption and partitioning processes

$$R = 1 + \frac{\theta_w}{\theta_a K_H} + \frac{\rho_b K_{Dsat}}{\theta_a K_H} + \frac{K_{IA} A_{IA}}{\theta_a} \qquad (A11)$$

where $\theta_w$ is volumetric water content, $\theta_a$ is gas-filled porosity, $K_H$ (dimensionless) is Henry's constant, $K_{Dsat}$ (cm$^3$ g$^{-1}$) is the sorption coefficient for water-saturated conditions, $\rho_b$ (g cm$^{-3}$) is the dry soil bulk density, $K_{IA}$ (cm) is the adsorption coefficient between gas and the gas-water interface and $A_{IA}$ (cm$^2$ cm$^{-3}$) is the specific surface area of the gas-water interface. The terms on the right-hand side describe retardation by the gas phase (1), partitioning into the aqueous phase (2), adsorption on the grains (3) and the last term accounts for gas-water interfacial adsorption (4).

## Appendix B:  Detailed experimental results

Tables 5 and 6 show the theoretical migration velocity, the injection duration (slug), the injected mass, and the normalized recovery of the components $CS_2$ and argon. The tables list all experiments in order according to the conducted series.

*Author contributions.*  Simon M. Kleinknecht designed and conducted this experimental study. Holger Class and Jürgen Braun were responsible for the scientific and experimental supervision. Simon M. Kleinknecht prepared the manuscript with contributions from both co-authors.

The authors declare that they have no conflict of interest.





**Table 5.** Experimental conditions of vapor-retardation experiments in fine glass beads: series, experiment, theoretical seepage velocity, injection duration, and injected mass and recovery of $CS_2$ and argon.

| Series # | Exp. # | v cm h$^{-1}$ | $t_{inj}$ h | $m_{Ar}$ mg | $m_{CS2}$ mg | $r_{Ar}$ - | $r_{CS2}$ - |
|---|---|---|---|---|---|---|---|
| Fine glass beads | | | | | | | |
| 1 | 1 | 25 | 27.80 | 667.8 | 2671.4 | 0.994 | 1.123 |
| | 2 | 50 | 13.92 | 677.7 | 2710.9 | 1.011 | 1.066 |
| | 3 | 50 | 14.10 | 675.9 | 2703.5 | 0.996 | 1.005 |
| | 4 | 50 | 13.65 | 677.7 | 542.2 | 0.992 | 1.033 |
| | 5 | 50 | 14.29 | 690.3 | 13.8 | 0.988 | 0.822 |
| | 6 | 50 | 14.07 | 687.0 | 13.7 | 0.984 | 0.753 |
| | 7 | 50 | 13.82 | 673.3 | 13.5 | 0.987 | 0.770 |
| 2 | 8 | 25 | 28.35 | 619.9 | 12.4 | 0.989 | 1.022 |
| | 9 | 50 | 14.40 | 617.8 | 12.4 | 0.993 | 0.983 |
| | 10 | 50 | 12.83 | 615.2 | 12.3 | 0.997 | 1.010 |
| | 11 | 100 | 7.22 | 620.8 | 12.4 | 1.000 | 0.941 |
| | 12 | 200 | 3.56 | 617.1 | 12.3 | 0.998 | 0.941 |
| 3 | 13 | 50 | 14.38 | 635.0 | 12.7 | 0.995 | 1.054 |
| | 14 | 50 | 14.22 | 634.0 | 12.7 | 1.000 | 1.032 |
| | 15 | 50 | 14.43 | 635.4 | 12.7 | 0.989 | 1.013 |
| 4 | 16 | 50 | 14.06 | 626.0 | 12.5 | 0.994 | 1.039 |





**Table 6.** Experimental conditions of vapor-retardation experiments in Geba fine sand: series, experiment, theoretical seepage velocity, injection duration, and injected mass and recovery of $CS_2$ and argon.

| Series # | Exp. # | $v$ cm h$^{-1}$ | $t_{inj}$ h | $m_{Ar}$ mg | $m_{CS2}$ mg | $r_{Ar}$ - | $r_{CS2}$ - |
|---|---|---|---|---|---|---|---|
| Geba fine sand | | | | | | | |
| 1 | 17 | 25 | 28.71 | 673.9 | 13.5 | 0.982 | 0.964 |
| | 18 | 25 | 40.97 | 944.5 | 18.9 | 0.992 | 0.942 |
| | 19 | 50 | 14.30 | 599.2 | 12.0 | 0.985 | 1.006 |
| | 20 | 50 | 14.31 | 597.5 | 12.0 | 0.983 | 0.970 |
| | 21 | 100 | 7.20 | 664.8 | 13.3 | 0.999 | 0.994 |
| | 22 | 100 | 7.32 | 665.9 | 13.3 | 0.999 | 1.062 |
| | 23 | 200 | 3.83 | 706.2 | 14.1 | 1.000 | 0.994 |
| | 24 | 200 | 3.54 | 657.7 | 13.2 | 0.995 | 0.962 |
| 2 | 25 | 25 | 41.17 | 795.7 | 15.9 | 0.984 | 1.009 |
| | 26 | 50 | 14.33 | 552.8 | 11.1 | 1.000 | 0.955 |
| | 27 | 50 | 20.28 | 767.3 | 15.3 | 0.992 | 1.115 |
| | 28 | 50 | 20.62 | 799.5 | 16.0 | 0.997 | 0.973 |
| | 29 | 100 | 10.12 | 797.6 | 16.0 | 0.987 | 0.965 |
| | 30 | 200 | 4.67 | 800.8 | 16.0 | 1.000 | 1.013 |
| 3 | 31 | 50 | 20.54 | 815.8 | 16.3 | 0.994 | 0.854 |
| | 32 | 50 | 24.04 | 955.8 | 19.1 | 0.998 | 0.684 |
| | 33 | 100 | 11.02 | 864.1 | 17.3 | 1.000 | 0.534 |
| | 34 | 200 | 5.16 | 782.9 | 15.7 | 1.008 | 0.689 |
| | 35 | 25 | 49.26 | 957.3 | 19.1 | 1.012 | 0.010 |
| | 36 | 100 | 10.28 | 793.0 | 15.9 | 1.000 | 0.174 |
| | 37 | 50 | 23.81 | 944.0 | 18.9 | 0.996 | 0.016 |





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
