# Peer review of "Experimental study on retardation of a heavy NAPL vapor in partially saturated porous media"

_Hydrology and Earth System Sciences, 2016_

## Referee Comment (RC1) · Anonymous Referee #1 · 27 Sep 2016

General Comments The paper describes experiments designed to measure CS2 vapor transport/retardation in dry and moist unsaturated porous media, and generally does so in a satisfactory manner. The paper does not provide a compelling motivation for these experiments, however. The introduction briefly discusses the importance of understanding gravity effects on dense vapors, because of the preferential downward flow (i.e., toward aquifers) of such vapors. Density-driven flow is never mentioned again, however; it is neither treated theoretically in the handling of the experimental data nor are the experiments themselves (unlike the first author's other published work)1 designed to examine density-driven flow (the experiments examine upward advection, rather than downward density-facilitated flow). Thus, the first stated motivation

for studying CS2 vapor transport is left unaddressed, leaving only the vague call for "further (contaminant-specific) investigations. . .to improve the process understanding required to assess the threat to the environment." Although the experiments appear to be generally well executed, the paper does not deliver on elucidating the contaminant-specific processes that influence CS2 retardation.

Rather than determining which processes affect CS2 retention, they made relatively unsupported simplifying assumptions that a single process is important (only dissolution) and then chalked up the discrepancy between their theoretical and measured retardation to the processes they had previously excluded without actually investigating the nature, magnitude, or feasibility of these other processes to close the gap. More specifically, despite its potential importance (which is acknowledged), the authors excluded theoretical treatment of the process of accumulation at the air-water interface because "determining the size of the interfacial areas and partitioning parameters in theoretical approaches is considered a challenge". The most recent paper they cite regarding this challenge is 18 years old. I encourage the authors to consult more recent literature regarding air-water interfacial area determination and solute accumulation, including those by some of the authors they cite elsewhere (e.g., Brusseau, Costanza-Robinson)2-9 and others (e.g., Wildenschild, Kibbey)10-13 Costanza-Robinson et al.9 for example, provide an empirical relationship for sandy materials describing the associated between porous media surface area, moisture saturation, and air-water interfacial areas that could be tailored to the media examined here. With this relationship or others, the authors could assess the potential for interfacial accumulation to feasibly account for the additional retardation experienced most notably on the fine sand.

The authors also summarily excluded sorption of CS2 to the solid phase from consideration because of its supposed (no numbers are provided) similarity to CO2 with regard to air-water partitioning and solubility. The correlation the authors are drawing between solubility and/or air-water partitioning and the solid-phase sorption coefficient is unclear. Sorption to the solid phase is more typically correlated to a compound's

octanol-water partitioning constant (KOW), which certainly differs substantially for CO2 and CS2. Moreover, a saturated phase experiment could readily be conducted and KD measured rather than making such an assumption.

Specific Recommendations a) A revised introduction that provides a more compelling motivation for studying CS2 retardation and a more accurate framing of the experimental work to follow would allow readers to recognize what aspects of the work are novel and scientifically significant. For example, if CS2 retardation is really truly understudied (a quick search in a well established database revealed very few CS2 papers, which, coupled with its prevalence at NPL sites, surprised me) than say so. At several points in the paper, the authors refer to their experimental setup as "novel" (including conclusion #1), but basis of this claim is unclear; what exactly is novel about the setup and what processes/variables/systems does it open to investigation that were previously precluded?

b) Conduct a saturated phase experiment to measure KD and, if not available in the literature, a surface tension experiment to measure KIW before excluding sorption at solid and interfacial phases from consideration.

c) As appropriate, use parameters from (b) in your theoretical model of CS2 retardation to perform a more complete and rigorous process-based analysis of your experimental findings.

d) The significance of the dispersion/dispersivity parameters derived for CS2/the porous media is not clear to the reader. If you are going to perform this analysis, what is the important take-home for readers? As it stands, several of the conclusions are underwhelming – moments analysis works (conclusion #2), simple theoretical constructs from 1961 are imperfect (conclusion #3), diffusion effects increase with longer residence time (conclusion #4).

e) The conclusion most directly tied to the goal of the paper and potentially of greater interest to readers is #5, but suffers from interpretations based on assumptions of what

processes control transport, since those processes were not specifically studied. The experiments and analysis suggested above in (a) would strengthen the conclusions that could be drawn.

f) Conclusion #6 is potentially quite interesting, but needs more discussion and incorporation of more relevant literature. I accept that further experimental investigation of the biodegradation may lie beyond the scope of the current paper, but if the data are going to be presented at all, they should be discussed (e.g., the feasibility of anaerobic degradation to occur at such timescales; if CS2 is degraded so thoroughly so quickly (recoveries of only 1%!) then why has CS2 persisted at so many of the NPL sites for so long? etc.)

g) Conclusion #7 is a bit disorganized, repeating some of #5 and #6 before recommending that SVE be used for CS2 remediation. This recommendation could be elaborated upon by discussion of what is actually being done and with what degree of success at the many NPL sites contaminated with CS2. Also, some caveat should be included, given that the volatility and rate of evaporation of CS2 liquid was not studied.

Technical Points a) If I understand the intended meaning correctly, "irreducible saturation" is more typically termed "residual saturation" b) I didn't understand the concept of filling the porous media columns "each with an overfill of around 30 cm". c) I found the schematic of the experimental system to be overly detailed to the point that it limited reader comprehension. I believe the He tank should be Ar instead? Several items in the figure weren't in the legend. The purpose of the Tedlar bags was not clear. I would dramatically simply the figure. d) The rationale for bottom-up flow was never provided and seems to counter the stated motivation of examining density-driven flow. e) The paragraph containing lines 1-10 on p 6 seemed particularly disorganized, jumping around from the N2 chase to the gas mixture, back to the chase. f) Although 7 experiments are described (series 1-4 for glass beads; 1-3 for fine sand), only a fraction of these had full data – no saturation profiles for 3 of the 7; and poor mass recovery for series 3 fine sand. Because the saturation was at the heart of arguments regarding

CS2 retardation, the missing saturation profiles for these experiments is noteworthy. That said, the accuracy of the saturation profiles was called into question on p 9. The validity of basing arguments on profiles that are simultaneous dismissed as misleading due to the small size of the tensiometers was confusing. Moreoever, column mass had been measurement throughout the experiment and supposedly provided an independent measure of moisture saturation that was more reliable. Why weren't these data shown instead of the tensiometer data (e.g., in Figure 3)? I don't mean to imply you should only show data you agree with, but if you fundamentally do believe that the tensiometer data are inaccurate, why present them to readers? g) As the authors note, it is not uncommon for compound-specific behavior to get lumped into dispersivity values, and also common for dispersivity values for nonreactive tracers to be considered more reliable. The authors might therefore consider using the non-reactive tracer data to arrive at a dispersivity value and fix this as an input parameter in the dispersion fitting of the CS2. h) The authors repeatedly mention grain-size distribution as a variable potentially influencing retardation. Presumably some of the grain-size effect is through its relationship to surface area (and therefore would affect solid-phase sorption and air-water interfacial accumulation). Some discussion and theoretical handling of the surface area impacts on retardation is needed.

1. Kleinknecht, S. M.; Class, H.; Braun, J. Density-driven migration of heavy NAPL vapor in the unsaturated zone. Vadose Zone J 2015, 14, 10.102136/vzj2014.12.0173. 2. Brusseau, M. L.; El Ouni, A.; Araujo, J. B.; Zhong, H. Novel methods for measuring air-water interfacial area in unsaturated porous media. Chemosphere 2015, 127, 208-201310.1016/j.chemosphere.2015.01.029. 3. Brusseau, M. L.; Janousek, H.; Murao, A.; Schnaar, G. Synchrotron X-ray Microtomography and Interfacial Partitioning Tracer Test Measurements of NAPL-water Interfacial Areas. Water Resources Research 2008, 44, W0141110.1029/2006WR005517. 4. Brusseau, M. L.; Peng, S.; Schnaar, G.; Costanza-Robinson, M. S. Relationships among air-water interfacial area, capillary pressure, and water saturation for a sandy porous medium. Water Resources Research 2006, 42, W03501. 5. Brusseau, M. L.; Peng, S.; Schnaar, G.;

[Figure]
Murao, A. Measuring air-water interfacial areas with X-ray microtomography and interfacial partitioning tracer tests. Environmental Science and Technology 2007, 41, 1956-61. 6. Costanza, M. S.; Brusseau, M. L. Contaminant vapor adsorption at the gas-water interface in soils. Environmental Science and Technology 2000, 34, 1-11. 7. Costanza-Robinson, M. S.; Brusseau, M. L. Air-water interfacial areas in unsaturated soils: Evaluation of interfacial domains. Water Resources Research 2002, 38, 1195-211. 8. Costanza-Robinson, M. S.; Carlson, T. D.; Brusseau, M. L. Vapor-phase transport of trichloroethene in an intermediate-scale vadose-zone system: Retention processes and partitioning-tracer-based prediction. Journal of Contaminant Hydrology 2013, 145, 82-8910.1016/j.jconhyd.2012.12.004. 9. Costanza-Robinson, M. S.; Harrold, K. H.; Lieb-Lappen, R. M. X-ray microtomography determination of air-water interfacial area-water saturation relationships in sandy porous media. Environmental Science and Technology 2008, 42, 2949-5610.1021/es072080d. 10. Culligan, K. A.; Wildenschild, D.; Christensen, B. S. B.; Gray, W. G.; Rivers, M. L.; Tompson, A. F. B. Interfacial area measurements for unsaturated flow through a porous medium. Water Resources Research 2004, 40, W12413W12413 10.1029/2004wr003278. 11. Chen, L.; Kibbey, T. C. G. Measurement of air-water interfacial area for multiple hysteretic drainage curves in an unsaturated fine sand. Langmuir 2006, 22, 6874-80. 12. Chen, L.; Miller, G. A.; Kibbey, T. C. G. Rapid pseudo-static measurement of hysteretic capillary pressure-saturation relationships in unconsolidated porous media. Geotechnical Testing Journal 2007, 30, 474-82. 13. Kibbey, T. C. G.; Chen, L. A pore network model study of the fluid-fluid interfacial areas measured by dynamic-interface tracer depletion and miscible displacement water phase advective tracer methods. Water Resources Research 2012, 48, W1051910.1029/2012WR011862.

---

## Short Comment (SC1) · 30 Sep 2016

We would like to thank you for this thorough and descriptive review. We agree that parts of the manuscript lack clarity and need to be revised to make the objective and results of our work better accessible and coherent.

We concur that the main weak point of our manuscript is the simplifying assumption that only dissolution is responsible for the retention of CS2 while we neglected adsorption at the air-water interface and at the solid phase. Our intention for a revised manuscript is to address this issue and consult recent publications (as indicated) regarding these processes as well as look into the feasibility to conduct own batch experiments under saturated conditions to determine the missing sorption coefficient KD. This appears

to us as the key issue to provide a revised manuscript with improved and more well-grounded conclusions on transport and retardation of CS2.

Moreover, we plan to look more carefully into the process and timescales of biodegradation, though to our knowledge only few publications are available. Unfortunately, experiments to quantify biodegradation lie beyond the scope of this manuscript.

Other issues raised in the review will be addressed by providing a better motivation and elaborating the reasoning more precisely. Your comments in this review will greatly help us to do so.

---

## Referee Comment (RC2) · Anonymous Referee #2 · 6 Nov 2016

General Comments:

This manuscript investigates the transport of carbon disulfide (CS2) vapor in partially saturated porous media. Certainly, the experimental design is interesting, and the experimental data presented deserve to be published. However, the analysis of the results presented is not very thorough. Furthermore, the manuscript is neither organized nor written very well.

Specific Comments:

(1) An important concern with this work is that, although in a gaseous state is 1.6 times denser that air, density effects have not been considered in the estimation of the

retardation factor.

(2) The authors must clearly describe the novel contributions of this study. Vapor retardation due to partitioning into the aqueous phase is an intuitive result.

Technical Corrections:

(1) Page 1, line 10, ". . . as a function of porous medium . . ." is awkward.

(2) The legend of Figure 1 is incomplete. Some of the apparatus are not listed.

(3) The are many repetitions in the manuscript. For example, the authors have mentioned several times throughout the manuscript that the experiments were conducted in two different porous media (fine glass beads and Geba fine sand) under both dry and partially saturated (moist) conditions.

(4) The manuscript should be checked very carefully for grammatical errors. For example, (page 7, line 10) "an separate"; (page 7, line 29) insert "the" before "Henry's."

(5) All symbols must be defined in the manuscript as soon as they appear. For example, none of the symbols in equation (1) have been defined.

(6) The first sentence in the "Results and discussion" section does not belong there. It is more of introductory material.

(7) Figure 9 deserves more attention. It is hard to follow.

———————————

---

## Author Comment (AC1) · 17 Nov 2016

**1   Comments**

We thank the reviewers for their careful observations and important comments. We agree that by providing a revised introduction and conclusion, adding a thorough discussion of all potential contributions to retardation of CS2, and addressing the technical issues raised will improve the quality of our manuscript and hope to get an invitation to do so.

[Figure]

**1.1   Rewriting the introduction and conclusions**

In a revision, we will change the introduction to give a more compelling motivation for our work in the context of vapor transport in the unsaturated zone, to clarify the reference to density-driven migration, and to underpin the need for the contaminant-specific investigation of CS2. Changing the introduction and addressing the issues/questions will provide a more accurate framing and will emphasize the scientifically important aspects of our work (Review #1: comment a). The conclusions will be restructured focusing on the important findings of our work, the transport processes and retardation of CS2. Minor conclusions such as the discussion of dispersion/dispersivity parameters (Review #1: comment d) will be merged. Effects of density difference on retardation (Review #2: comment 1) were not a goal of our experiments, which aimed at a clear process differentiation between retardation and density-driven migration. Nevertheless, experiments were conducted with bottom-up as well as downward oriented flow showing no impact on the retardation of CS2. Moreover, we will provide a better discussion (Review #1: comment f and g) regarding biodegradation of CS2 (conclusion #6 ) with input from recent publications as well as our recommendation for soil-vapor extraction (conclusion #7).

**1.2   Discussion of potential contribution to retardation of CS2**

From a scientific point of view the "retardation issue" is the most critical one (Review #1: comment b and c, Review #2: comment 2). This refers to the "relatively unsupported simplifying assumptions" that adsorption on the solid phase and at the air–water interface may be neglected "because of its supposed similarity to $CO_2$ with regard to air–water partitioning and solubility". We have since carefully consulted publications on the two neglected contributions to retardation. The findings and new insights regarding conclusion #5 (Review #1: comment e) will be included in a revision of the manuscript:

[Figure]

**Sorption of a compound on the solid phase** is governed by the partitioning co-efficient $K_D$ of that particular compound. The coefficient for CS2 can be estimated with $K_D = K_{OC} \times f_{OC}$, where $K_{OC}$ (L kg$^{-1}$) is the soil organic carbon partitioning coefficient and $f_{OC}$ the fraction of organic carbon in the soil material. For CS2, the coefficient $K_{OC}$ is 45.7 L kg$^{-1}$ according to the Superfund Soil Screening Guidance, US EPA (1996). Howard (1990) report that "Carbon disulfide in solution would there-fore not be expected to adsorb significantly to soil" due to the relatively low $K_{OC}$. In our manuscript we introduced the two types of materials used, fine glass beads and Geba fine sand, with a d50 of 162 and 140 um determined from particle-size anal-ysis (Mastersizer 2000, Malvern Instruments Ltd. Worcestershire, United Kingdom). The chemical composition of fine glass beads is given by the manufacturer (Sigmund Lindner GmbH, SiLibeads Typ S 100..200 um) as SiO2 (72.5 %), Na2O (13 %), CaO (9.1 %), MgO (4.2 %) , and Al2O3 (0.58 %) which is soda-lime glass. Geba fine sand (Quarzsande GmbH, Geba weiss, 63..350 um) is composed of SiO2 (99.2 %), Fe2O3 (0.09 %), Al2O3 (1.85 %), and TiO2 (0.24 %), thus a pure quartz sand. Both materi-als contain negligible fractions of organic carbon ($f_{OC}$) supporting our assumption that sorption of CS2 on the solid phase in our experiments does not significantly contribute to retardation and, hence, may be neglected.

**Adsorption on the air–water interface** in a partially water-saturated porous medium depends on the air–water interfacial area $A_{IA}$ and the air–water partitioning coefficient $K_{IA}$. Since we do not have the technical equipment to directly measure the air–water interfacial area in our experiment columns, we used the correlation (Eq. 1) proposed by AUTHOR (YEAR) to estimate it. This correlation is based on X-ray microtomography measurements of glass beads and natural sands.

$$A_{IA} = SA[(-0.9112)S_W + 0.9031] \tag{1}$$

where $SA$ is the geometric surface area according to the smooth-sphere assumption ($SA_{GBfine} = 22.77\ cm^{-1}$ and $SA_{Geba} = 24.32\ cm^{-1}$). Figure 1 shows the air–water interfacial areas as a function of water saturation of our materials: fine glass beads

**Fig. 1.** Calculated air–water interfacial area as a function of water saturation with correlation proposed by AUTHOR (YEAR) for materials used in experiments.

(GBfine) and Geba fine sand. We estimated the air–water partitioning coefficient $K_{IA}$ of CS2 using the empirical correlation proposed by Valsaraj (1988). He found a correlation (Eq. 2) between the interfacial-water partitioning constant $K_{IW}$ ($K_{IW} = K_H K_{IA}$) and the octanol–water partitioning coefficient $K_{OW}$.

$$K_{IW} = 3 \times 10^{-7} K_{OW}^{0.68} \tag{2}$$

This correlation with $log(K_{OW}) = 2.00$ (US EPA , 1996) and the dimensionless Henry coefficient $K_H = 1.04$ (Lide , 1996) yielded the air–water partitioning coefficient of CS2 $K_{IA} = 6.87 \times 10^{-6} cm$ . The contribution of dissolution into the bulk water and of adsorption to the air–water interfacial to the theoretical retardation coefficient (Eq. 3) can calculated using the following equation:

$$R_t = 1 + \beta_w + \beta_{IA} = 1 + \frac{\theta_w}{\theta_a K_H} + \frac{K_{IA} A_{IA}}{\theta_a}. \tag{3}$$

Figure 2 shows the retardation coefficient with air–water interfacial adsorption (green line with triangles) and without adsorption (black line). Since the difference of estimated $A_{IA}$ between fine glass beads and Geba fine is marginal, only one theoretical coefficient is plotted in Fig. 2. The ratio between the contributions from dissolution $\beta_w$ ($\frac{\theta_w}{\theta_a K_H}$) and from air–water interfacial adsorption $\beta_{IA}$ ($\frac{K_{IA} A_{IA}}{\theta_a}$) at $S_w = 0.162$ yields $\beta_w/\beta_{IA} = 513$. While these calculations indicate that our assumption to neglect adsorption on grains and air–water interface is correct, they cannot account for the deviation between the retardation coefficient measured for Geba fine sand at $S_w = 0.162$ and the corresponding theoretical value. Since dissolution, which is a function of the Henry coefficient and assumes equilibrium, is already at its maximum and adsorption on the grains may be neglected as discussed above, it is postulated that this increased

**Fig. 2.** Updated graph of retardation coefficients including contribution of air–water interfacial adsorption in theoretical retardation coefficient (green line with triangles).

**Fig. 3.** Initial water saturation profiles in fine glass beads and Geba fine sand.

retardation can only be caused by air–water interfacial adsorption. Possible reasons for an underestimation of the adsorption to the air–water interface could be that the inhomogeneous water saturation profile (see Fig. 3), the grain-size distribution, or an underestimation of the specific surface of the Geba fine sand particles lead to an underestimation of the interfacial area. Figure 4 and 5 suggest that the smooth-sphere assumption holds for glass beads but not for Geba fine sand, thus the interfacial area for Geba fine sand might have been significantly underestimated. Moreover, interfacial areas measured with microtomography are significantly smaller than those measured with gas phase tracer experiments (Brusseau , 2006; AUTHOR, YEAR). Costanza (2000) observed the maximum interfacial area for water saturation in the range of 15 to 25 % of water saturation. They also reported the possibility of multilayer adsorption and that the actual adsorption may be significantly underestimated when true $A_{IA}$ values are used.

Ascribing the observed discrepancy in Geba fine sand to air–water interfacial adsorption of CS2, an interfacial area of about $A_{IA,calc} = 6553\ cm^{-1}$ would be required to obtain retardation factor of $R_{Geba} = 1.31$ (Eq. 3). This is consistent with measured interfacial areas from vapor-phase tracer experiments by Costanza (2000).

In conclusion, the contributions to the retardation based on the parameters, coefficients, and correlations introduced above have shown that for glass beads sorption on

**Fig. 4.** SEM pictures of fine glass beads.

**Fig. 5.** SEM pictures of Geba fine sand.

the solid phase and on the air–water interfacial area does not significantly contribute to retardation of CS2. An interfacial area of about two orders of magnitude higher would be required to account for the discrepancy observed between the experimental and theoretical coefficient in Geba fine sand. Such areas have been found in vapor-phase tracer experiments but not with microtomography measurements, indicating that the $A_{IA}$ values based on microtomography may not be applicable for the evaluation of retardation on partially water-saturated Geba fine sand. Additional investigations with selected partitioning tracers might yield a better understanding of the different contribution of air–water interfacial adsorption to retardation in fine glass beads compared to Geba fine sand. Nevertheless, we believe that the experiments shown and the (additional) conclusions drawn are conclusive and will substantially strengthen conclusion #5.

**2 Technical points/issues**

**Review #1:**

1. We use "irreducible saturation" for water and "residual saturation" for NAPLs. If this is misleading we will replace the term by "residual saturation".

2. The concept of filling the columns will be re-written to improve comprehensibility.

3. The flowchart of the experimental set-up will be simplified and corrected.

4. The choice fell on bottom-up flow due to technical reasons and since no effect of downward oriented flow on retardation has been observed.

5. We will revise the discussion about water saturation and tensiometer measurements. We do trust the measurements at steady state (equilibrium) to derive water saturation profiles prior to the experiments but we are carefully about tensiometer measurements during ongoing vapor injection.

6. We will look into the experimental dispersivity values and their validity and significance.

7. We discussed the impacts of surface area on retardation in the detailed argumentation above. Measurements of specific surface area with BET were below the detection limit of our device.

**Review #2:**

1. We will revise our statement.

2. Figure 1 will be simplified and corrected.
3. We will carefully revise the manuscript to avoid redundancy

4. and grammatical errors.

5. Correct placement and definitions of mathematical symbols will be checked.

6. First paragraph of "Results and discussion" will be revised.

7. We will provide a more detailed description of Figure 9 and may simplify it.

**References**

Brusseau, M. L., Peng, S., Schnaar, G., and Costanza-Robinson, M. S.: Relationships among air-water interfacial area, capillary pressure, and water saturation for a sandy porous medium, Water Resources Research, 42, n/a–n/a, doi:10.1029/2005WR004058, http://dx.doi.org/ 10.1029/2005WR004058, w03501, 2006.

Costanza, M. S., and Brusseau, M. L.: Contaminant Vapor Adsorption at the Gas–Water Interface in Soils, Environmental Science Technology, 34, http://pubs.acs.org/doi/abs/10.1021/es9904585, 2000.

Costanza-Robinson, M. S., Harrold, K. H., and Lieb-Lappen, R. M.: X-ray Microtomography Determination of Air–Water Interfacial Area-Water Saturation Relationships in Sandy Porous Media, Environ. Sci. Technol., 42, 2949–2956, doi:10.1021/es072080d, http://pubs.acs. org/doi/abs/10.1021/es072080d, 2008.

Howard, P., Sage, G., Jarvis,W., and Gray, D.: Handbook of environmental fate and exposure data for organic chemicals. Volume II: Solvents, Chelsea, MI (US); Lewis Publishers, Inc., 1990.

Lide, D. R.: CRC Handbook of chemistry and physics, CRC Press, 86 edn., 2005.

US Environmental Protection Agency: Superfund Soil Screening Guidance – Technical Background Document – Part 5: Chemical-Specific Parameters, Tech. rep., Office of Solid Waste and Emergency Response, https://semspub.epa.gov/work/HQ/175235.pdf, 1996.

Valsaraj, K. T.: On the physico-chemical aspects of partitioning of non-polar hydrophobic organics at the air-water interface, Chemosphere, 17, 875 – 887, doi:http://dx.doi.org/10.1016/0045-6535(88)90060-4, http://www.sciencedirect.com/science/article/pii/ 0045653588900604, 1988.
* * *
[Figure]

**Fig. 6.** Calculated air–water interfacial area as a function of water saturation with correlation proposed by Costanza-Robinson (2008) for materials used in experiments.

Fig. 7. Updated graph of retardation coefficients including contribution of air–water interfacial adsorption in theoretical retardation coefficient (green line with triangles).

[Figure]

[Figure]

**Fig. 8.** Initial water saturation profiles in fine glass beads and Geba fine sand.

[Figure]

**Fig. 9.** SEM pictures of fine glass beads.

[Figure]

**Fig. 10.** SEM pictures of Geba fine sand.

---

## Author Response (AR1)

**Point-by-point reply to reviews on manuscript "Experimental study on retardation of a heavy NAPL vapor in partially saturated porous media"**

**Anonymous Referee #1**

| Reference | Content |
|---|---|
| **Point a)** | **A revised introduction that provides a more compelling motivation for studying CS2 retardation and a more accurate framing of the experimental work to follow would allow readers to recognize what aspects of the work are novel and scientifically significant. For example, if CS2 retardation is really truly understudied (a quick search in a well established database revealed very few CS2 papers, which, coupled with its prevalence at NPL sites, surprised me) than say so. At several points in the paper, the authors refer to their experimental setup as "novel" (including conclusion #1), but basis of this claim is unclear; what exactly is novel about the setup and what processes/variables/systems does it open to investigation that were previously precluded?** |
| **Reply** | We have thoroughly revised the introduction incorporating recent research papers. We clarified the objective of our study, and improved the reasoning, in particular to show why studying retardation of CS2 is important. We are confident that the revised introduction will frame our experimental work more accurately and will be easier for readers to follow. |
| **Point b)** | **Conduct a saturated phase experiment to measure KD and, if not available in the literature, a surface tension experiment to measure KIW before excluding sorption at solid and interfacial phases from consideration.** |
| **Reply** | We have carefully consulted, among others, the papers indicated in your review and have come to the conclusion that sorption on the solid phase may be neglected based on the properties of the porous media and the physicochemical properties of CS2. A more detailed discussion is provided in Section 3.3 Retardation of CS2 on p. 19. |
| **Point c)** | **As appropriate, use parameters from (b) in your theoretical model of CS2 retardation to perform a more complete and rigorous process-based analysis of your experimental findings.** |
| **Reply** | See reply to Point b) |
| **Point d)** | **The significance of the dispersion/dispersivity parameters derived for CS2/the porous media is not clear to the reader. If you are going to perform this analysis, what is the important take-home for readers? As it stands, several of the conclusions are underwhelming – moments analysis works (conclusion #2), simple theoretical constructs from 1961 are imperfect (conclusion #3), diffusion effects increase with longer residence time (conclusion #4).** |
| **Reply** | We agree that the discussion of the dispersion/dispersivity parameters and the corresponding conclusions can be improved. We decided to shorten the discussion significantly and to focus on more important findings and conclusions concerning retardation and partitioning processes. |
| **Point e)** | **The conclusion most directly tied to the goal of the paper and potentially of greater interest to readers is #5, but suffers from interpretations based on assumptions of what processes control transport, since those processes were not specifically studied. The experiments and analysis suggested above in (a) would strengthen the conclusions that could be drawn.** |
| **Reply** | We have reconsidered our assumptions of which processes control transport, we |

carefully consulted relevant papers, and we incorporated a detailed discussion in "Section 3.3 Retardation of CS2" of our manuscript. We trust that the revised discussion of the partitioning processes improves the paper as a whole; moreover it supports the quality of the interpretation of our experimental results.

| | |
|---|---|
| **Point f)** | **Conclusion #6 is potentially quite interesting, but needs more discussion and incorporation of more relevant literature. I accept that further experimental investigation of the biodegradation may lie beyond the scope of the current paper, but if the data are going to be presented at all, they should be discussed (e.g., the feasibility of anaerobic degradation to occur at such timescales; if CS2 is degraded so thoroughly so quickly (recoveries of only 1%!) then why has CS2 persisted at so many of the NPL sites for so long? etc.)** |
| **Reply** | We thank the reviewer for encouraging the discussion related to biodegradation of CS2. We have carefully consulted published studies which are all (except for Cox et al., 2013, see reference list in manuscript) concerned with waste-gas treatment plants using biofilters for manufacturing companies. They have found a small number of microbes capable of oxidizing CS2 in aerobic or anaerobic conditions. However, they also reported self-inhibitory effects with increasing CS2 concentrations. Hence, biodegradation of CS2 is only relevant in specific environments and under conditions which are most likely not met close to source zones at CS2-contaminated sites. We have incorporated the discussion at the end of "Section 3.3 Retardation of CS2" where the mass recovery is discussed; we also revised our conclusion. A comparison of our results with reported degradation rate constants from batch experiments was not feasible since effluent concentration measurements at the column outflow are the only available data and do not allow for calculations of rate constants. |
| **Point g)** | **Conclusion #7 is a bit disorganized, repeating some of #5 and #6 before recommending that SVE be used for CS2 remediation. This recommendation could be elaborated upon by discussion of what is actually being done and with what degree of success at the many NPL sites contaminated with CS2. Also, some caveat should be included, given that the volatility and rate of evaporation of CS2 liquid was not studied.** |
| **Reply** | We have revised the conclusion to improve our reasoning. SVE is the method of choice for the remediation of VOCs in the unsaturated zone and applies especially for CS2 given its physicochemical properties. Of course, SVE cannot be readily applied to the saturated zone when a spill of CS2 (DNAPL) reaches and penetrates the groundwater. It has to be combined with, for instance, steam injection or soil venting to actively vaporize contaminants. |
| **Technical point a)** | **If I understand the intended meaning correctly, "irreducible saturation" is more typically termed "residual saturation".** |
| **Reply** | We generally used "irreducible" for water and "residual" for NAPL. We now consistently use residual throughout the manuscript to avoid misunderstandings. |
| **Technical point b)** | **I didn't understand the concept of filling the porous media columns "each with an overfill of around 30 cm".** |
| **Reply** | The concept of filling the columns was rephrased to improve comprehensibility. |
| **Technical point c)** | **I found the schematic of the experimental system to be overly detailed to the point that it limited reader comprehension. I believe the He tank should be Ar instead? Several items in the figure weren't in the legend. The purpose of the Tedlar bags was not clear. I would dramatically simply the figure.** |
| **Reply** | The flow chart of the experiment has been re-designed and simplified. |
| **Technical point d)** | **The rationale for bottom-up flow was never provided and seems to counter the stated motivation of examining density-driven flow.** |
| **Reply** | These experiments aimed at providing a basic understanding of vapor retardation with a clear differentiation from density-driven flow. Bottom-up flow was chosen since the injected vapor is heavier than soil air. Stable flow conditions (e.g. reduce fingering) could be ensured and additional influences (e.g. gravity) could be avoided. Density-driven flow |

was investigated in an earlier experimental investigation.

| | |
|---|---|
| **Technical point e)** | **The paragraph containing lines 1-10 on p 6 seemed particularly disorganized, jumping around from the N2 chase to the gas mixture, back to the chase.** |
| **Reply** | The paragraph has been revised and shortened. |
| **Technical point f)** | **Although 7 experiments are described (series 1-4 for glass beads; 1-3 for fine sand), only a fraction of these had full data – no saturation profiles for 3 of the 7; and poor mass recovery for series 3 fine sand. Because the saturation was at the heart of arguments regarding CS2 retardation, the missing saturation profiles for these experiments is noteworthy. That said, the accuracy of the saturation profiles was called into question on p 9. The validity of basing arguments on profiles that are simultaneous dismissed as misleading due to the small size of the tensiometers was confusing. Moreoever, column mass had been measurement throughout the experiment and supposedly provided an independent measure of moisture saturation that was more reliable. Why weren't these data shown instead of the tensiometer data (e.g., in Figure 3)? I don't mean to imply you should only show data you agree with, but if you fundamentally do believe that the tensiometer data are inaccurate, why present them to readers?** |
| **Reply** | We have revised the discussion about the water saturation profiles and tensiometer measurements since it was obviously prone to confusion. The missing water saturation profiles of Series 2 and 3 in fine glass beads are mentioned in the manuscript. We trust the tensiometer measurement at static conditions, but we question the measurement quality during active gas flow, especially with respect to the fluctuations measured. Since we monitored the total column mass by placing the entire set-up on a scale, we could assure that a drying-out of the porous medium was prevented. However, actually showing this data (basically a constant value) does not add to the manuscript. Hence, we decided to remove the right-hand graph of Figure 3 and only show the measured water saturation profiles. |
| **Technical point g)** | **As the authors note, it is not uncommon for compound-specific behavior to get lumped into dispersivity values, and also common for dispersivity values for nonreactive tracers to be considered more reliable. The authors might therefore consider using the non-reactive tracer data to arrive at a dispersivity value and fix this as an input parameter in the dispersion fitting of the CS2.** |
| **Reply** | Thank you for this suggestion. We did consider this approach and believe that it might give us a slightly different perspective, but overall, this will not improve the interpretation of our data. |
| **Technical point h)** | **The authors repeatedly mention grain-size distribution as a variable potentially influencing retardation. Presumably some of the grain-size effect is through its relationship to surface area (and therefore would affect solid-phase sorption and air-water interfacial accumulation). Some discussion and theoretical handling of the surface area impacts on retardation is needed.** |
| **Reply** | We have provided a detailed discussion in "Section 3.3 Retardation of CS2" related to the partitioning processes and their dependence on soil material and chemical compound properties. We are confident that the newly added information will improve our manuscript. |

**Anonymous Referee #2**

| Reference | Content |
| --- | --- |
| Point 1 | **An important concern with this work is that, although in a gaseous state is 1.6 times denser that air, density effects have not been considered in the estimation of the retardation factor.** |
| Reply | Partitioning processes leading to retardation are not expected to be influenced by density effects, but vapor migration indeed is. Impact of density difference on vapor plume migration (advection) was investigated in a separate study (see Kleinknecht (2015) in reference list of manuscript). See also our reply to Technical Point d) above. |
| Point 2 | **The authors must clearly describe the novel contributions of this study. Vapor retardation due to partitioning into the aqueous phase is an intuitive result.** |
| Reply | We have thoroughly revised our manuscript to highlight the novel contribution of our study. There are no experimental studies on retardation of CS2 vapor available. Hence, our study provides new data on this subject which also has been made available (see remark on data availability in manuscript). |
| Technical point 1 | Page 1, line 10, "... as a function of porous medium ..." is awkward. |
| Reply | Revised. |
| Technical point 2 | The legend of Figure 1 is incomplete. Some of the apparatus are not listed. |
| Reply | Figure 1 has been re-designed and simplified. |
| Technical point 3 | The are many repetitions in the manuscript. For example, the authors have mentioned several times throughout the manuscript that the experiments were conducted in two different porous media (fine glass beads and Geba fine sand) under both dry and partially saturated (moist) conditions. |
| Reply | We have carefully checked the manuscript and removed redundant phrases. |
| Technical point 4 | The manuscript should be checked very carefully for grammatical errors. For example, (page 7, line 10) "an separate"; (page 7, line 29) insert "the" before "Henry's." |
| Reply | Thank you for bringing this to our attention. |
| Technical point 5 | All symbols must be defined in the manuscript as soon as they appear. For example, none of the symbols in equation (1) have been defined. |
| Reply | We have checked the symbols and their correct definitions. |
| Technical point 6 | The first sentence in the "Results and discussion" section does not belong there. It is more of introductory material. |
| Reply | Revised. |
| Technical point 7 | Figure 9 deserves more attention. It is hard to follow. |
| Reply | We have removed the velocity-dependent symbol size and revised the entire section to facilitate reading. |

**List of relevant changes**

1. Revision of abstract.

2. Revision of introduction.

3. Shortening and revision of section „Materials and methods".

    1. Replaced Figure 1 by a simplified version of the flowchart.

    2. Included scanning electron microscopy images of porous media in „Materials and methods".

4. Revision of section „Data evaluation"; including now theoretical approaches to estimate air-water interfacial adsorption.

5. Revision of section „Results and discussion".

    1. Shortening of subsection „3.1 Water saturations"; removed right-hand side graph showing tensiometer recordings over time in Figure 4.

    2. Shortening of subsection „3.2 Impact of velocity on breakthrough"; removed Figure 6 and Figure 7.

    3. Major revision of subsection „3.3 Retardation of CS2"; added thorough discussion about partitioning processes and revised paragraph about biodegradation.

6. Major revision of conclusions including newly added insights.

[revised manuscript text omitted]